# Emergence of planar cell polarity from the interplay of local interactions and global gradients

**Divyoj Singh[1†], Sriram Ramaswamy[2], Mohit Kumar Jolly[1]\*, Mohd Suhail Rizvi[3]\***

[1]Department of Bioengineering, Indian Institute of Science, Bangalore, India; [2]Centre for Condensed Matter Theory, Department of Physics, Indian Institute of Science, Bengalore, India; [3]Department of Biomedical Engineering, Indian Institute of Technology, Hyderabad, India

**Abstract** Planar cell polarity (PCP) – tissue-scale alignment of the direction of asymmetric localization of proteins at the cell-cell interface – is essential for embryonic development and physiological functions. Abnormalities in PCP can result in developmental imperfections, including neural tube closure defects and misaligned hair follicles. Decoding the mechanisms responsible for PCP establishment and maintenance remains a fundamental open question. While the roles of various molecules – broadly classified into 'global' and 'local' modules – have been well-studied, their necessity and sufficiency in explaining PCP and connecting their perturbations to experimentally observed patterns have not been examined. Here, we develop a minimal model that captures the proposed features of PCP establishment – a global tissue-level gradient and local asymmetric distribution of protein complexes. The proposed model suggests that while polarity can emerge without a gradient, the gradient not only acts as a global cue but also increases the robustness of PCP against stochastic perturbations. We also recapitulated and quantified the experimentally observed features of swirling patterns and domineering non-autonomy, using only three free model parameters - rate of protein binding to membrane, the concentration of PCP proteins, and the gradient steepness. We explain how self-stabilizing asymmetric protein localizations in the presence of tissue-level gradient can lead to robust PCP patterns and reveal minimal design principles for a polarized system.

**\*For correspondence:**
mkjolly@iisc.ac.in (MKJ);
suhailr@bme.iith.ac.in (MHR)

**Present address:** †Department of Physics, UC Santa Barbara, United States

## Editor's evaluation

This study presents a valuable model for the emergence of planar cell polarity from the interplay of local interactions and global gradient. The framework of this model is solid. A quality of this model is its simplicity and its convenience for experimental testing.

## Introduction

Epithelial tissues form the outermost layer of most organs including skin. They not only function as a protective layer for the inner tissues but also take part in several physiological functions including absorption, secretion, excretion and filtration (*Torras et al., 2018*). Epithelial cells usually manifest two major types of polarity: *Apico-basal polarity* is asymmetry in the distribution of cellular components and functions between the two surfaces of an epithelial sheet (*Buckley and St Johnston, 2022*) and *planar cell polarity* (PCP) refers to organization in a direction parallel to the plane of the epithelial sheet (*Butler and Wallingford, 2017*). PCP is an evolutionarily conserved phenomenon that is required for early developmental events such as neural tube closure and later in the formation of functional organs (*Papakrivopoulou et al., 2021*). It has been a topic of extensive scientific investigations in several

model organisms, including *Drosophila* and vertebrates (*Maung and Jenny, 2011*; *Jones and Chen, 2007*; *Wansleeben and Meijlink, 2011*). In these studies, PCP has been investigated to understand its molecular mechanisms and the fallouts of its disruptions in the physiology of several organs, such as hair bristles on the wing, abdomen, and ommatidia in *Drosophila* (*Strutt and Strutt, 2021*; *Strutt and Strutt, 2002*; *Ambegaonkar and Irvine, 2015*), and inner ear and epidermis in mammals (*Simons and Mlodzik, 2008*; *Dong et al., 2018*; *Figure 1A, B*).

Disruption in PCP can lead to a variety of developmental abnormalities such as neural tube defects, idiopathic scoliosis, skeletal dysplasia manifested as short limbs and craniofacial anomalies (*Butler and Wallingford, 2017*) and tracheal ciliary malfunction (*Vladar et al., 2012*). PCP is also essential for the auditory and vestibular function of the vertebrate inner ear. It determines how mechanosensory hair cells sense sound waves (*Deans, 2021*) with PCP defects resulting in hearing loss (*Li et al., 2021*). Therefore, understanding the multi-scale dynamics of PCP establishment and maintenance is a question of fundamental importance.

The molecular players involved in the establishment of PCP in various organisms have been well studied (see *Strutt and Strutt, 2021* for a comprehensive review of the topic). Although the interactions among the proteins involved in PCP are quite intricate and not very precisely demarcated, they have been broadly categorized into two groups, also known as *modules*: Global and Core (*Butler and Wallingford, 2017*).

The global module contains two atypical cadherins, Fat (Ft) and Dachsous (Ds), the kinase Four-jointed (Fj), and the myosin Dachs (D) (*Mao et al., 2011*). Ft and Ds from adjacent cells interact with each other to form heterodimers at the cell-cell interface. Their affinity for each other can be modulated by Fj (*Brittle et al., 2010*; *Ishikawa et al., 2008*; *Hale et al., 2015*). Dachs co-localises with Ds and generates active forces at the apical end, thus regulating organ shape (*Matis and Axelrod, 2013*). Proteins of global modules are usually present in tissue-level expression gradients (*Lawrence and Casal, 2018*; *Gao et al., 2011*). These tissue-level gradients are known to provide the global cue to PCP machinery, which helps alignment of the PCP with the axis of the organ (*Strutt and Strutt, 2021*).

We emphasize, however, that despite the nomenclature, the mechanisms of the interactions among the proteins of the global module are also based on intercellular protein interactions. These intercellular interactions are usually in the form of Ft-Ds heterodimers which are, at the cellular level, local in nature.

On the other hand, the core or local module consists of Frizzled (Fz), Flamingo (Fmi), Van Gogh (Vang), Prickle-spiny-legs (Pk), Dishevelled (Dsh) and Diego (Dgo) (*Yang and Mlodzik, 2015*) which interact with one another, some intercellularly and some intracellularly, by forming complexes. Similar to Ft-Ds heterodimers across a cell-cell boundary, Fmi from two neighbouring cells form complexes. The trans-membrane protein Fz recruits the cytosolic proteins Dsh and Dgo, and localises in proximity of Fmi in one of the two cells. Another trans-membrane protein Vang recruits cytosolic protein Pk close to Fmi on the other cell. These interactions within the core module have been argued to be sufficient for establishing a local order (polarity in two adjacent cells; *Butler and Wallingford, 2017*; *Devenport, 2014*).

The mechanistic understanding of PCP crucially requires establishing the relative contributions of global agencies such as gradients and local interactions like heterodimer formation in the emergence and maintenance of polarity (*Lawrence et al., 2007*; *Fisher et al., 2019*). Most theoretical and computational models have considered the role of the global interactions to be limited to only providing directional cues (*Aw and Devenport, 2017*). In principle, global fields such as tissue-level gradient can of course help *maintain* polarity in the presence of perturbations but, if intercellular interactions are weak, they can also *create* polarity to an extent proportional to the gradient magnitude. Strong enough local interactions can cooperatively give rise to a strong susceptibility for macroscopic (at the scale of the tissue) polarity. A global field can then, therefore, produce a disproportionate tissue-scale polarization. How redundant or synergistic the roles of the two features – asymmetric protein localization and tissue-level gradients – remains to be decoded.

Given the complex interactions between molecules within and across modules, mathematical modelling has been an essential tool to account for experimental abnormalities and make further testable experimental predictions. Despite the two-dimensional nature of epithelial tissue, most mathematical models of the PCP have been formulated for one dimension (e.g. *Jolly et al., 2014*; *Mani*

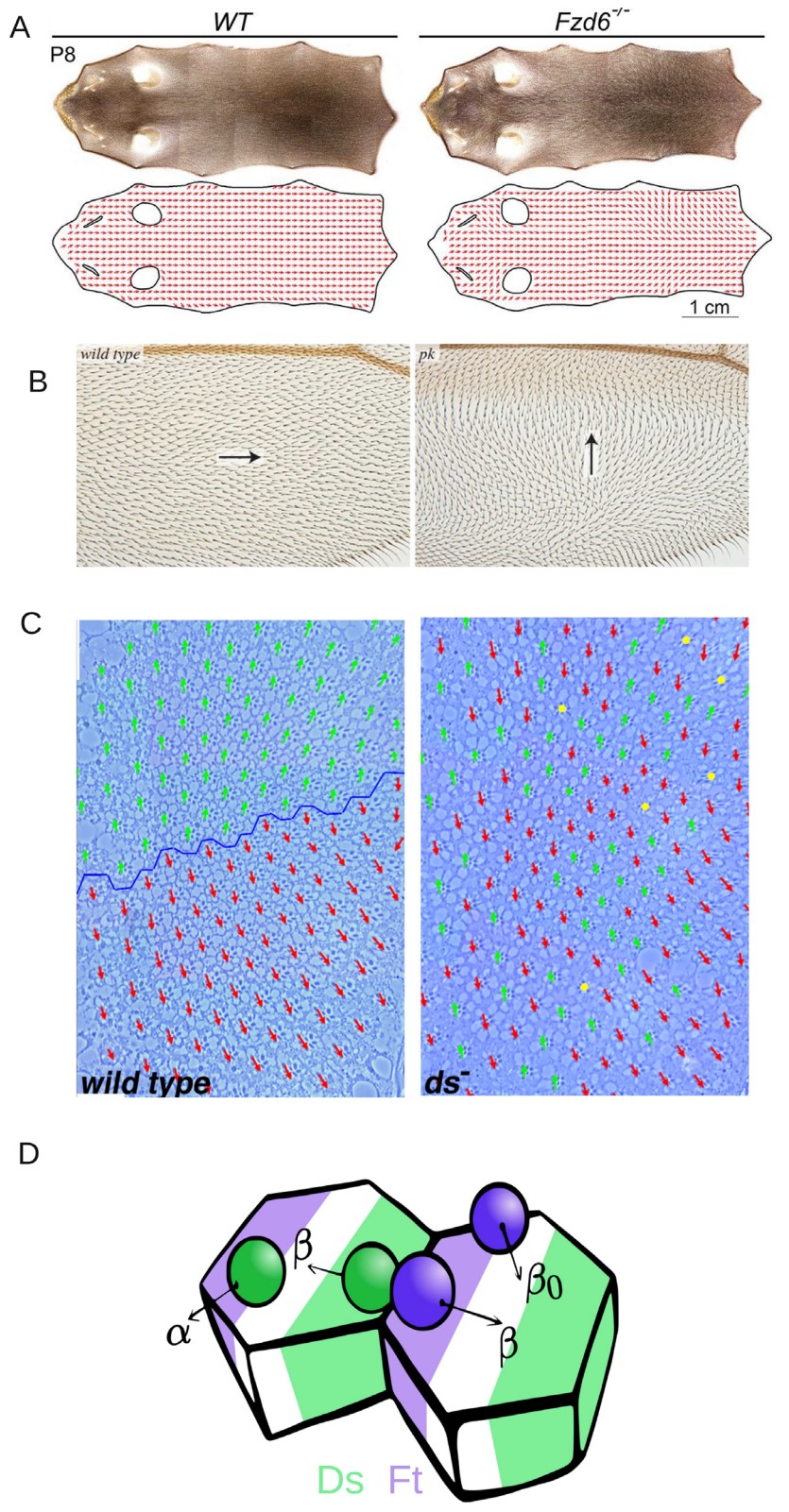

**Figure 1.** Introduction to planar cell polarity and the model. Examples of PCP and its disruption in (**A**) mouse skin (reproduced from Figure 3A of **Dong et al., 2018**), (**B**) *Drosophila melanogaster* wing (reproduced from Figure 4G and H of **Ambegaonkar and Irvine, 2015**), and (**C**) *Drosophila melanogaster* eye leading to irregular ommatidial arrangement (reproduced from Figure S1A and B of **Simon, 2004**). In (**C**), green and red arrows

*Figure 1 continued on next page*

*Figure 1 continued*

represent ommatidial orientation. The blue line indicates the equator. (**D**) Schematic depicting the protein binding, unbinding kinetics, and heterodimer formation in a two-cell system. Ft (purple bead) and Ds (green bead) bind to the membrane with rate $\alpha$ and unbind with rate $\beta_0$. The unbinding rate is reduced to $\beta$ if a heterodimer is formed by the presence of both proteins on the membrane. The green and purple shaded regions indicate higher concentrations of Ft and Ds at the cell membrane, respectively.

*et al., 2013*; *Fisher and Strutt, 2019*; *Schamberg et al., 2010*), with exceptions where the biological question studied required a two-dimensional framework (*Burak and Shraiman, 2009*; *Amonlird-viman et al., 2005*; *Hazelwood and Hancock, 2013*; *Shadkhoo and Mani, 2019*). There are broadly two classes of models for PCP. One class takes into account the mechanistic details of the system and explains examples of PCP establishment and its disruption in specific organs (see *Axelrod and Tomlin, 2011* for review). While they provide useful inputs for the experimental contexts for which they are formulated, their generalizability across parameter sets or biological contexts is limited. The other class are phenomenological models that apply to a wider variety of cases and parameter ranges because they ignore some mechanistic details but account for the pattern formation (*Hazelwood and Hancock, 2013*).

Here, we integrate the strengths of both these kinds of models in the context of the global module of PCP. We incorporate minimal mechanistic relations with the goal of capturing the general dynamical properties of the system, together with a focus on specific interesting experimental observations. Towards this, we focus on the intercellular interactions of Ft and Ds to form heterodimer complexes and look at the effect of heterodimer formation on the asymmetric localization of these proteins on cell membranes. We have developed a minimal model of the asymmetric enrichment of Ft and Ds in one and two-dimensional frameworks with only three free parameters. At the microscopic level, we consider the interaction between two proteins at the cell-cell interface, in contrast to other works in which a presumed intercellular interaction between intercellular heterodimers is considered (*Burak and Shraiman, 2009*; *Mani et al., 2013*).

With this minimal model, we recapitulate some of the known PCP results, such as the establishment of PCP without any global cue (*Devenport, 2014*). We further demonstrate the dual role of the tissue-level expression gradients in PCP establishment and its coordination with the tissue axis. We also test the robustness of the PCP in the minimal model in the presence of static and dynamic noises representing cell-to-cell variations in expression levels of the proteins and thermal fluctuations in protein binding kinetics, respectively. Finally, we also explain non-homogeneous PCP patterns on deleting specific proteins from a region and make predictions that can be tested experimentally.

## Mathematical modeling of planar cell polarity
### A short review of other modeling approaches
As mentioned earlier, there are two broad classes of modeling approaches for PCP dynamics. In models with mechanistic details, the concentrations of specific proteins on the cell membranes are estimated from mass-action-based kinetics. In *Amonlirdviman et al., 2005*; *Fisher et al., 2019*; *Fisher and Strutt, 2019*; *Hale et al., 2015*; *Jolly et al., 2014*, the PCP proteins are considered to bind to each other across the membranes of the neighboring cells to form protein complexes (homo-/heterodimers in case of core and global modules, respectively) with a protein-protein interaction based feedback mechanism. The cell polarity is then described in terms of the asymmetric localization of intercellular complexes.These mechanistic models involve a very large parameter space which makes it difficult to demarcate the roles of specific interactions in PCP establishment and maintenance.

In the semi-phenomenological approaches (*Burak and Shraiman, 2009*; *Mani et al., 2013*), the mechanism of feedback is considered in terms of the interactions between protein complexes at the intracellular and intercellular levels, in contrast to the interaction between individual proteins. The protein complex of one polarity is assumed to inhibit (promote) the formation of a complex of opposite (same) polarity. Even though these models reproduce experimental findings and make testable predictions, the mechanistic basis for the assumption of interactions at the level of protein complexes has not been verified experimentally.

In the phenomenological approach (*Hazelwood and Hancock, 2013*), the PCP is studied based on models of magnetization where the global cue is also integrated akin to an external magnetic field. These models do not involve many parameters but also do not reveal any mechanistic insights about the PCP.

Here, we are proposing a minimal model for PCP to show that only two proteins are sufficient to establish PCP via intercellular interactions which has also been seen in in vitro experiments (*Loza et al., 2017*). Another major deviation of the proposed model from the previous works is defining cell polarity in terms of asymmetry in the protein concentration and not of the inter-cellular protein complexes. Experimentally, it is not too difficult to precisely quantify the subcellular distributions of proteins. This makes the proposed definition of cell polarity more practical as compared to that in terms of heterodimers which are often measured via co-localization (a proxy measure) of the participating proteins.

## Basic assumptions

We assume epithelial tissue to be a non-motile confluent layer of cells. For simplicity, we ignore the three-dimensional organ-specific epithelial architecture and assume identical apical-basal geometry of the cells and a flat planar tissue, and hence a purely two-dimensional description, which is adequate for describing planar cell polarity. This is a fair simplification and has remained a common feature across all the PCP modeling approaches (*Burak and Shraiman, 2009*; *Hazelwood and Hancock, 2013*; *Amonlirdviman et al., 2005*). In order to keep the analysis limited to PCP, we also do not take into account any cellular rearrangements, cell movements, or remodeling in our model. This simplification is based on the assumption that we are focusing on the late stages of PCP establishment, where cellular rearrangements have already taken place.

We assume that the protein translation and degradation happens at a very slow rate (large time scale). Therefore, each cell has a finite pool of the PCP proteins. For protein transport, we assume fast protein diffusion (small time scale) inside the cells. The PCP dynamics is considered to take place at a timescale somewhere between these two.

## Protein binding and transport kinetics

In order to build the model, we focus on the two members of the global module of PCP, Fat (or Ft) and Dachsous (or Ds), atypical cadherins that interact across the membranes of two adjacent cells (intercellular interaction). Both Ft and Ds, if present in the cytoplasm, can bind to the cell membrane and, if bound to the cell membrane, can detach and enter the cytoplasm. We assume within our model that in an 'isolated' cell (a hypothetical situation where intercellular interactions are ignored), the attachment and detachment of these cadherins follow the law of mass action, and the kinetic rates of attachment and detachment are $\alpha$ and $\beta_0$, respectively. We make the further simplifying assumption that these rate constants are the same for both proteins, but the argument can easily be generalized to disparate rate coefficients.

Finally, we take intercellular protein interaction at the cell membrane into account in the following manner. These cadherin proteins have been shown to form Ft-Ds heterodimers across cell membranes. Therefore, when a Ft (or Ds) is bound to the cell membrane, its rate of detachment goes down due to its interaction with the Ds (or Ft) from a neighboring cell (*Figure 1*). The modified rate of detachment can be written using Hill's function (*Alon, 2019*) as

$$\beta(c) = \beta_0 \times \mathcal{H}_2(c) = \frac{\beta_0}{1 + \left(\gamma c\right)^k} \tag{1}$$

where $c$ is the concentration of Ds or Ft, as the case may be, at the membrane of the neighboring cell, and $k$ is the cooperativity constant. This decrease in the detachment rate can be justified from the

observations of stabilization of Ft and Ds on the membrane due to their interactions with respective partners from the neighboring cells (*Hale et al., 2015*; *Loza et al., 2017*). In the context of the core module of PCP, similar mechanisms of Fmi stabilization on one cell membrane due to the presence of Fmi-Fz on the neighboring cell membrane have been recently shown (*Strutt et al., 2023*).

With this simple biochemical kinetic assumption, we now move on to the formulation of the dynamics of PCP in the tissue. First, we will describe it for a one dimensional tissue.

## One-dimensional model

In the 1D model, we assume a single row of cells. The total concentrations of Ft and Ds in cell are held constant thanks to the assumption mentioned previously.

Since in 1D each cell has only two ends, the concentrations of membrane-bound Ft at either side of the cells are denoted by $f_l(i)$ (amount of Ft on the left edge of the cell at location $i$), $f_r(i)$ (amount of Ft on the right edge of the cell at location index $i$). Similarly, for Ds these quantities are denoted by $d_l(i)$ and $d_r(i)$.

The equations describing the space-time evolution of concentrations of membrane-bound proteins are

$$\dot{f_l}(i) = \underbrace{\alpha(f_T(i) - f_m(i))}_{\text{Binding to Cell membrane}} - \underbrace{\beta_0 f_l(i)\mathcal{H}_2(d_r(i-1))}_{\text{Unbinding from Cell membrane}} , \tag{2}$$

$$\dot{f_r}(i) = \alpha(f_T(i) - f_m(i)) - \beta_0 f_r(i)\mathcal{H}_2(d_l(i+1)) \tag{3}$$

$$\dot{d_l}(i) = \alpha(d_T(i) - d_m(i)) - \beta_0 d_l(i)\mathcal{H}_2(f_r(i-1)) \tag{4}$$

$$\dot{d_r}(i) = \alpha(d_T(i) - d_m(i)) - \beta_0 d_r(i)\mathcal{H}_2(f_l(i+1)), \tag{5}$$

where $f_m(i) = f_l(i) + f_r(i)$ and $d_m(i) = d_l(i) + d_r(i)$ are the total membrane-bound protein in the $i^{\text{th}}$ cell. Here, $(\dot{\cdot})$ signifies the time-derivative of the quantity. The two terms on the right-hand sides of the equations represent the rates of protein binding and unbinding (which is dependent on the protein concentrations of other protein in the neighboring cell), respectively. Here, $f_T(i)$ and $d_T(i)$ stand for the amount of total (cytoplasmic + membrane bound) Ft and Ds present in the cell and can be dependent on the cell position in the tissue. In this work, we define the cell polarity in terms of the degree of asymmetric localization of the two proteins in the cell. We write the asymmetry or polarity of the two proteins in the $i^{\text{th}}$ cell as the difference between the protein concentrations on right and left edges of the cell, that is

$$p_f(i) = f_r(i) - f_l(i), \tag{6}$$

$$p_d(i) = d_r(i) - d_l(i), \tag{7}$$

Finally, using these definitions of protein polarity of individual cells, we define the overall tissue polarity as

$$p = \langle p(i) \rangle_i = \langle p_f(i) - p_d(i) \rangle_i, \tag{8}$$

where $\langle \cdot \rangle_i$ denotes the average over all the cells in the tissue. Please note that in this work $p$ (without cell index) denotes the polarization in the tissue (as obtained from *Equation 8*), whereas $p(i)$ (with cell index) signifies the polarity of the $i^{\text{th}}$ cell.

## Effect of noise in the system

Noise is ubiquitous in biological systems. The role of noise in PCP, therefore, is of utmost importance. This has led to some works in this direction using semi-phenomenological approaches (*Burak and Shraiman, 2009*) by considering noise only at the level of protein kinetics. Here, we have considered two sources of noise in the PCP as described below.

### Noise in protein kinetics

Following a similar approach as that in *Burak and Shraiman, 2009*, we also study the effect of noise due to protein kinetics by adding stochastic terms in the equations. Note: We do not expect our *one-dimensional* models with noise included to yield PCP in the limit of infinite system size, but they will give rise to coherent polarization on scales that grow as the noise intensity is decreased. For the

stochastic simulations for the 1D case, we modify the equations by adding zero-mean Gaussian white noise terms of small magnitudes, leading to the Langevin equations

$$\dot{f}_l(i) = \alpha(f_T(i) - f_m(i)) - \beta_0 f_l(i) \mathcal{H}_2(d_r(i-1)) + L_f(i,t) \tag{9}$$

$$\dot{f}_r(i) = \alpha(f_T(i) - f_m(i)) - \beta_0 f_r(i) \mathcal{H}_2(d_l(i+1)) + R_f(i,t) \tag{10}$$

$$\dot{d}_l(i) = \alpha(d_T(i) - d_m(i)) - \beta_0 d_l(i) \mathcal{H}_2(f_r(i-1)) + L_d(i,t) \tag{11}$$

$$\dot{f}_r(i) = \alpha(d_T(i) - d_m(i)) - \beta_0 d_r(i) \mathcal{H}_2(f_l(i+1)) + R_d(i,t) \tag{12}$$

where the noise terms such as $L_F(i,t)$ are uncorrelated in time $t$ and position $i$ of the cell in the tissue, and with respect to location $L$ and $R$ of the cell edges, and protein type $Ft$ and $Ds$, that is

$$\langle P_A(i,t_u)Q_B(i,t_v)\rangle \propto \delta_{ij}\delta(t_u - t_v)\delta_{PQ}\delta_{AB} \tag{13}$$

where $P$ and $Q$ can take values $L$ and $R$, $A$ and $B$ label the protein type (Ft or Ds), and the $\delta$ symbols are Dirac or Kronecker deltas depending on whether their argument is a continuous or discrete variable. In principle, the noise strengths depend on the protein concentrations. Our replacement of this 'multiplicative noise' by an additive noise can be justified *post facto* if we approximate the concentrations by their mean values. In order to implement this, we select a random number for each cell edge $T(i,P,A,t_u) \in \mathcal{N}(0,1)$ from a standard Normal distribution with mean $\mu = 0$ and standard deviation 1. Using this, we write the noise term in the equation as

$$P_A(i,t_u) \quad = \eta \times T(i,P,A,t_u) \tag{14}$$

where $\eta$ represents the amplitude of the noise or the magnitude of stochasticity in the system.

## Static spatial noise in total protein concentration of Ft and Ds

Further, even for a uniform tissue-level protein expression, no two cells are expected to have identical protein levels. Therefore, we have also considered small cell-to-cell variations in the total protein levels:

$$\begin{aligned} f_T(i) &= \bar{f}_T + S_f(i) \\ d_T(i) &= \bar{d}_T + S_d(i) \end{aligned} \tag{15}$$

where $\bar{f}_T$ and $\bar{d}_T$ are the average protein levels in the cells of the tissue and the noise terms $S_f(i), S_d(i)$ are uncorrelated along position $i$ of the cell in the tissue, that is

$$\langle S_A(i)S_B(j)\rangle \sim \delta_{ij}\delta_{AB} \tag{16}$$

where the labels $A$, $B$ can be $f$ or $d$, and $i$ and $j$ stand for the position of the cell. To implement noise in total protein concentration, we selected a random number $T(A,i) \in \mathcal{N}(0,1)$ from a Standard Normal distribution with mean 0 and standard deviation 1. Using this, we write the noise term in the equation as

$$S_A(i) \quad = S \times T(A,i) \tag{17}$$

where $S$ represents the amplitude of the noise or the magnitude of stochasticity in the system.

For both noise types, we present results for the polarization averaged over 50 realizations of each. For these many repetitions, we observe convergence in the tissue polarity averaged over repetitions.

## Two-dimensional model

For the 2D model, we used a lattice of immotile cells formed of regular hexagons with each cell having six neighbors as opposed to two in 1D. Following same approach as in 1D, we write equations for dynamics of protein concentration for a cell at $\mathbf{x}$

$$\dot{f}(\mathbf{x},\theta_i) = \underbrace{\alpha\left(f_T(\mathbf{x}) - f_m\right)}_{\text{Binding to Membrane}} - \underbrace{\beta_0 f(\mathbf{x},\theta_i)\mathcal{H}_2\left(d(\mathbf{x}+\mathbf{x}_{\theta_i},\theta_i+\pi)\right)}_{\text{Unbinding from Membrane}} + \underbrace{\mathcal{D}_\theta \frac{\partial^2}{\partial\theta^2}a(\mathbf{x},\theta,t)}_{\text{Diffusion along membrane}} \tag{18}$$

$$\dot{d}(\mathbf{x},\theta_i) = \alpha\left(d_T(\mathbf{x}) - d_m\right) - \beta_0 d(\mathbf{x},\theta_i)\mathcal{H}_2\left(f(\mathbf{x}+\mathbf{x}_{\theta_i},\theta_i+\pi)\right) + \mathcal{D}_\theta \frac{\partial^2}{\partial\theta^2}b(\mathbf{x},\theta,t) \tag{19}$$

where

$$f_m(\mathbf{x}, t) = \sum_{\theta_i} f(\mathbf{x}, \theta_i, t) \tag{20}$$

$$d_m(\mathbf{x}, t) = \sum_{\theta_i} d(\mathbf{x}, \theta_i, t) \tag{21}$$

are the total membrane-bound protein concentrations in the cell. Here we sum over the protein concentration at the 6 angles $\theta_i = 0, \pi/3, 2\pi/3, \pi, 4\pi/3$ and $5\pi/3$ (6 parts of discretised cell membrane). $f_T(\mathbf{x}, t)$ and $d_T(\mathbf{x}, t)$ are the total (cytoplasmic +membrane bound) protein levels in the cell located at $\mathbf{x}$, and $f(\mathbf{x} + \mathbf{x}_{\theta_i}, \theta_i + \pi, t)$ represents the protein Ft in the neighboring cell. Similar to the one-dimensional model, we can define the planar cell polarity (PCP) of a protein in a cell in terms of the asymmetric distribution of that protein on the cell membrane, leading to the vectors

$$\mathbf{p}_f(\mathbf{x}, t) = \sum_{\theta_i} f(\mathbf{x}, \theta_i, t) \begin{pmatrix} \cos\theta_i \\ \sin\theta_i \end{pmatrix} \tag{22}$$

$$\mathbf{p}_d(\mathbf{x}, t) = \sum_{\theta_i} d(\mathbf{x}, \theta_i, t) \begin{pmatrix} \cos\theta_i \\ \sin\theta_i \end{pmatrix}. \tag{23}$$

characterizing the local orientational asymmetry in the distributions of Ft and Ds. Finally, we define

$$\mathbf{p} = \langle \mathbf{p}_f(\mathbf{x}) - \mathbf{p}_d(\mathbf{x}) \rangle_{\mathbf{x}} \tag{24}$$

as the tissue polarity where the average has been taken over the whole tissue.

## Continuum limit of the one-dimensional model

In 1D we can describe the protein distributions by $f_l(x)$ (amount of Ft on left edge of cell), $f_r(x)$, $d_l(x)$ and $d_r(x)$. Here, $x$ is the cell coordinate. The equations are ($x$ (position) and $t$ (time) dependencies are not written explicitly)

$$\dot{f}_l = \alpha(f_T - f_m) - f_l H_2(d_r^n) \tag{25}$$

$$\dot{f}_r = \alpha(f_T - f_m) - f_r H_2(d_l^n) \tag{26}$$

$$\dot{d}_l = \alpha(d_T - d_m) - d_l H_2(f_n^r) \tag{27}$$

$$\dot{d}_r = \alpha(d_T - d_m) - d_r H_2(f_l^n) \tag{28}$$

where $f_l(x), f_r(x), d_l(x), d_r(x)$ represent the amount of Ft and Ds and localized on the left or right edge of the cell, $d_n^r = d_r - \ell d_r', d_l^n = d_l + \ell d_l', f_r^n = f_r \ell f_r'$ and $f_l^n = f_l + \ell f_l'$ represent the concentration of Ds and Ft in the neighboring cell edges. ()' and (·) denote the spatial and time derivatives respectively, and $f_l(x), f_r(x), d_l(x), d_r(x)$ (total membrane-bound Ft in a cell at location $x$) and $d_m = d_l + d_r$. The terms $f_T$ and $d_T$ respectively stand for the amount of total (cytoplasmic + membrane-bound) Ft and Ds proteins in the cell and can be dependent on $x$, the cell location in the tissue.

We can define $p_f = f_r - f_l$ and $p_d = d_r - d_l$ to be the asymmetry (or polarity) in the localization of Ft and Ds on cells edges. Using the above equations, we can write the equations for $f_m, d_m, p_f$, and $p_d$ as

$$\begin{aligned}
\dot{f}_m = 2\alpha(f_T - f_m) &- \left(\frac{f_m - p_f}{2}\right) \mathcal{H}_2\left(\frac{d_m + p_d}{2},\right) \left[1 + k\left(\frac{d_m' + p_d'}{d_m + p_d}\right) \mathcal{H}_1\left(\frac{d_m + p_d}{2}\right)\right] \\
&- \left(\frac{f_m + p_f}{2}\right) \mathcal{H}_2\left(\frac{d_m - p_d}{2}\right) \left[1 - k\left(\frac{d_m' - p_d'}{d_m - p_d}\right) \mathcal{H}_1\left(\frac{d_m - p_d}{2}\right)\right]
\end{aligned} \tag{29}$$

$$\begin{aligned}
\dot{d}_m = 2\alpha(d_T - d_m) &- \left(\frac{d_m - p_d}{2}\right) \mathcal{H}_2\left(\frac{f_m + p_f}{2}\right) \left[1 + k\left(\frac{f_m' + p_f'}{f_m + p_f}\right) \mathcal{H}_1\left(\frac{f_m + p_f}{2}\right)\right] \\
&- \left(\frac{d_m + p_d}{2}\right) \mathcal{H}_2\left(\frac{f_m - p_f}{2}\right) \left[1 - k\left(\frac{f_m' - p_f'}{f_m - p_f}\right) \mathcal{H}_1\left(\frac{f_m - p_f}{2}\right)\right]
\end{aligned} \tag{30}$$

and

$$\dot{p}_f = \left(\frac{f_m + p_f}{2}\right) \mathcal{H}_2\left(\frac{d_m - p_d}{2},\right) \left[1 - k\left(\frac{d'_m - p'_d}{d_m - p_d}\right) \mathcal{H}_1\left(\frac{d_m - p_d}{2}\right)\right]$$
$$+ \left(\frac{f_m - p_f}{2}\right) \mathcal{H}_2\left(\frac{d_m + p_d}{2}\right) \left[1 + k\left(\frac{d'_m + p'_d}{d_m + p_d}\right) \mathcal{H}_1\left(\frac{d_m + p_d}{2}\right)\right] \tag{31}$$

$$\dot{p}_d = \left(\frac{d_m + p_d}{2}\right) \mathcal{H}_2\left(\frac{f_m - p_f}{2},\right) \left[1 - k\left(\frac{d'_m - p'_d}{d_m - p_d}\right) \mathcal{H}_1\left(\frac{d_m - p_d}{2}\right)\right]$$
$$+ \left(\frac{f_m - p_f}{2}\right) \mathcal{H}_2\left(\frac{d_m + p_d}{2}\right) \left[1 + k\left(\frac{d'_m + p'_d}{d_m + p_d}\right) \mathcal{H}_1\left(\frac{d_m + p_d}{2}\right)\right] \tag{32}$$

Here, we have assumed a weaks patial variation in the membrane-bound protein levels across the tissue which allows us to write

$$\mathcal{H}_2(a_r^n) \approx \mathcal{H}_2(a_r) \left(1 - k\mathcal{H}_1(a_r)\frac{a'_r}{a_r}\right) \tag{33}$$

$$\mathcal{H}_2(a_n^l) \approx \mathcal{H}_2(a_l) \left(1 - k\mathcal{H}_1(a_l)\frac{a'_l}{a_l}\right) \tag{34}$$

## Calculation of correlation length

In the presence of noise, the tissue need not be polarized in one direction. To estimate the length scale of the coordination in PCP, we calculate the correlation length of the polarization vector field. We start by calculating the correlation function of the polarization vector field at an instant of time. We use the following definition of the correlation function

$$C(r) = \frac{\sum_{ij} \mathbf{p}_i \cdot \mathbf{p}_j \delta\left(r_{ij} - r\right)}{\sum_{ij} \delta\left(r_{ij} - r\right)} \tag{35}$$

It tells us about the correlation between two vectors at a distance r. Next, we average this correlation function over time (for the last 300 time points out of 500). This helps in getting rid of variations in the correlation function due to transient behavior. Next, we integrate the area under the curve, which gives us the correlation length ($\zeta(p)$) of the polarization vector field.

## Parameter values

To understand the system analytically and reduce the modeling parameters, we non-dimensionalize the system using the following characteristic quantities

1. $\tau = 1/\beta_0$ for time
2. $1/\gamma$ for protein levels,

In the discrete tissue model the lengths are measured in terms of characteristic cell length. In the continuum limit of the model, the lengths are non-dimensionalized by the characteristic cell length, $\ell$. For the results shown in this manuscript we have set $k = 2$ but their qualitative nature does not depend on this specific choice. The non-dimensionalization results in just three free parameters in the system- Protein binding rate ($\alpha$), total protein concentrations $f_T$, and $d_T$. The protein concentrations are used as a control parameter to study the nature of polarization and the effects of protein gradient expressions, protein loss, and gain. These parameters, along with the initial conditions, can control the wide variety of patterns seen in PCP.

For the results without dynamic noise, the numerical simulations were performed until the change in the protein concentrations at each simulation time step drops blow $10^{-8}$. For the results with the dynamic noise the simulations were performed for a duration of 1.5 times (to ensure convergence) of that for their respective deterministic counterparts and the quantities were averaged over the last 1000 simulation steps.

To simulate the 1D model without dynamic noise, we considered an array of 500 cells and numerically solved the system of ordinary differential equations using the function solve_ivp from the SciPy package(version 1.6.2) in Python with adaptive time-stepping. For the 1D system with dynamic noise,

we performed all simulations using Euler's method with a fixed time step. The results showing average values of protein concentrations or polarity in the presence of dynamic or static noise the results were averaged over 50 realizations which was found to be sufficient for convergence in average and standard deviation values. For the 2D system, we take a hexagonal lattice of 50×50 cells and solve the equations for the Ft and Ds dynamics at cell boundaries using Euler's method.

In simulations, the tissues are initialized to have a very small amount of asymmetry in the protein distribution on the membranes (1% of total protein in the cell) to break the symmetry.

## Results
## Polarity can emerge even in the absence of gradient in protein expression

In PCP, tissue-level gradients of Ft and Ds are well documented (*Casal et al., 2006*; *Ma et al., 2003*; *Brittle et al., 2012*; *Ambegaonkar et al., 2012*), but their exact role and the mechanism of their action is not completely understood. Thus, we first ask whether polarity can emerge in the absence of any tissue-level gradient and noise in 1D model. Here for simplicity, we have kept the levels of Ft and Ds in each cell equal to $\rho$, referred to as total protein concentrations (membrane + cytoplasm), that is $f_T = d_T = \rho$. The results remain valid even if these levels are unequal. When $f_T \neq d_T$, the polarization magnitude of Ft and Ds become unequal.

We observe that with an increase in $\rho$, the total concentrations of membrane-bound Ft and Ds increase equally, that is $f_l + f_r = d_l + d_r \rho_m$. However, there is a threshold of total protein concentration, $\rho_c$, after which an increase in $\rho$ results in an abrupt but continuous increase in the membrane-bound protein levels (*Figure 2A*).

Below the threshold protein concentration, both proteins localize on both sides of the cell in equal amounts (*Figure 1D*). Therefore, there is no polarization of the cells (*Figure 2B and E* and *Figure 2—figure supplement 1*). But once we cross the threshold concentration, that is $\rho > \rho_c$; Ft gets bound to one side of the cell and Ds on the other side, resulting in non-zero polarization (*Figure 2B and E* and *Figure 2—figure supplement 1*). The initial conditions determine the direction of polarization here. The presence of a threshold is due to the inter-cellular interactions and heterodimer formation between the proteins across the membranes of two neighboring cells (*Figure 1* D). We note that the transition from an unpolarised state to a polarised state is a continuous one (*Figure 2—figure supplement 2*).

We also tested the role of the finite availability of the two proteins in each cell by replacing the term $\alpha(f_T(i) - f_m(i))$ (and similarly for Ds) with $\alpha f_\infty$, a constant factor (which effectively represents the unlimited pools of two proteins) in *Equations 2–5*. In this case, we found the protein distribution on the cell membrane to be symmetric in the steady state (*Figure 2—figure supplement 1C*). Next, we studied the dependence of the membrane-bound protein concentrations ($\rho_m$) as a function of the protein binding rate ($\alpha$) (*Figure 2C*).

Since the threshold protein concentration can depend on the binding rate $\alpha$, we generated a phase diagram of polarization as a function of protein binding rate, $\alpha$ and total protein concentration $\rho$ (*Figure 2D*). The phase diagram shows the existence of a polarized state ($p \neq 0$) and an unpolarised state ($p = 0$). The boundary separating the two states shows that with an increasing $\alpha$, the threshold concentration of proteins required for the polarization decreases.

To get an intuitive understanding of the establishment of polarization above the critical protein concentrations, we looked at the continuum limit of our discrete model of the 1D epithelium (*Equations 25; 30–32*). In the case of uniform protein expression levels, we get (see Appendix (Uniform protein expressions) for detailed derivation):

$$\dot{\rho}_m \approx 2\alpha(\rho - \rho_m) - \frac{4\rho_m}{\rho_m^2 + 4} \tag{36}$$

$$\dot{p} \approx -4p\left(\frac{4 - \rho_m^2}{(\rho_m^2 + 4)^2}\right) + \frac{p}{(\rho_m^2 + 4)^4}\left[A_p(\rho_m)p^2 + B_p(\rho_m)q^2\right] \tag{37}$$

where $q = p_f + p_d$, $p = p_f - p_d$ is the cell polarity as described in *Equation 8*, $\rho_m$ is the total concentration of the membrane-bound proteins. The exact forms of the functions $A_p(\rho_m)$ and $B_p(\rho_m)$ are

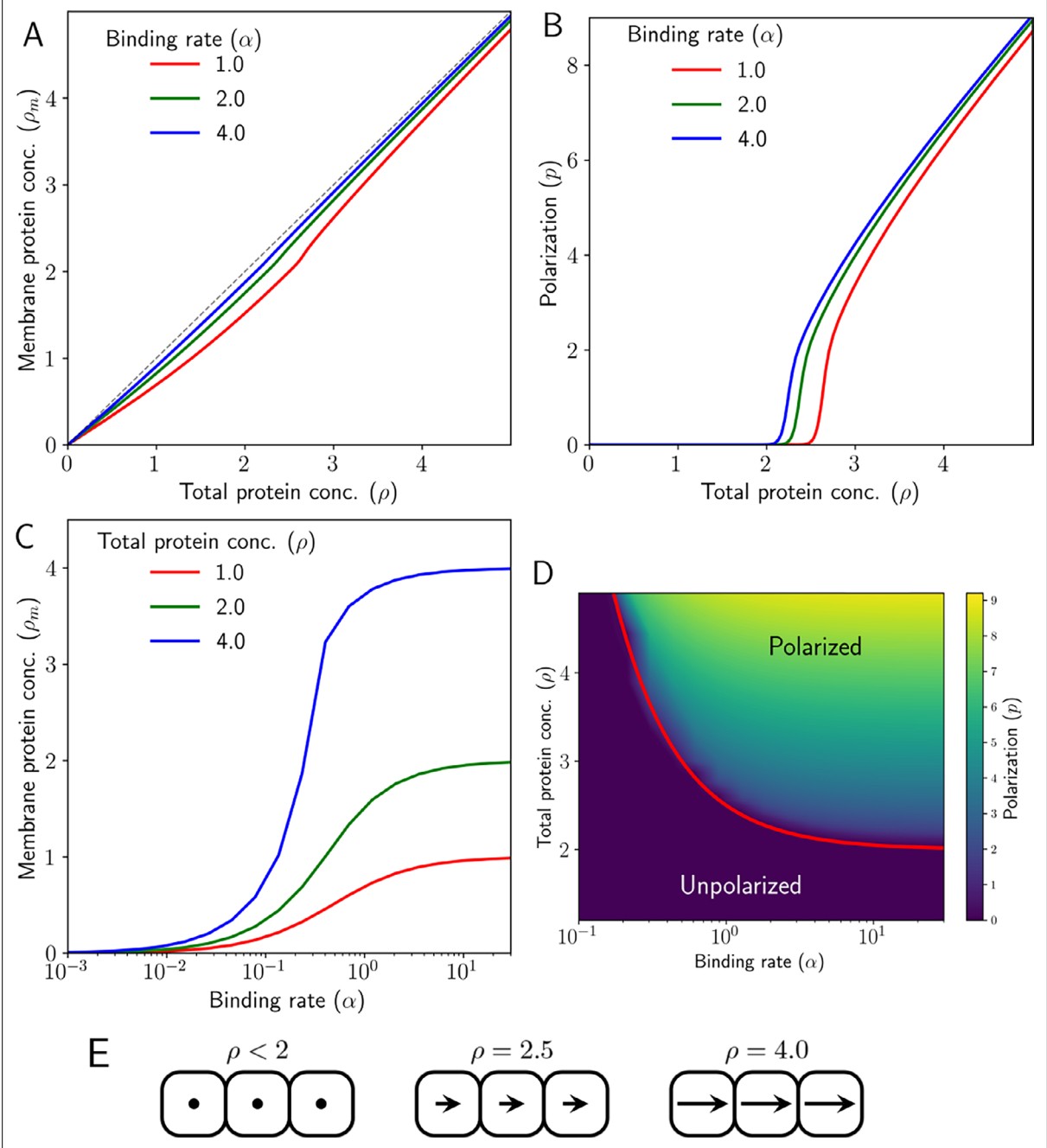

**Figure 2.** Emergence of polarity due to intercellular protein interactions in 1D. (**A**) Membrane protein concentration of Ft and Ds ($\rho_m$), and (**B**) polarization magnitude ($p$) at the steady state against total protein concentration ($\rho$). (**C**) Membrane protein concentration ($\rho_m$) at steady state as a function of protein binding rate ($\alpha$) at different total protein concentrations ($\rho$). (**D**) Phase diagram of polarization ($p$) as a function of $\rho\rho$ shows the existence of a polarized and unpolarised state. The red curve in (**D**) depicts the analytically obtained critical value in *Equation 29*. (**E**) Schematic representing cell polarization for different values of $\rho$. Here, arrows represent the direction and magnitude of cell polarity.

The online version of this article includes the following figure supplement(s) for figure 2:

**Figure supplement 1.** Dynamics in 1D: Level of Ft and Ds in the left and right side ($f_l, f_r, d_l$ and $d_r$) of the tissue as a function of time.

**Figure supplement 2.** Polarization magnitude as a function of $\Delta\rho = \rho - \rho_c$ for 1D.

specified in the Appendix. *Equation 37* shows that if $\rho_m > 2$, the state of zero cell polarity is unstable, and a nonzero value of polarization will set in. From *Equation 36*, we also find the critical protein level required for the polarization to be (marked in *Figure 2D*)

$$\rho_c = 2 + \frac{1}{2\alpha}. \tag{38}$$

This shows that the intercellular interactions, in the form of heterodimer formation between Ft and Ds, are sufficient for the emergence of polarity in the absence of any tissue-level expression gradient.

## Stochasticity results in partial loss of tissue level coordination in PCP

We have seen how spontaneous polarization of tissue can emerge in a completely deterministic treatment in which we model the dynamics of average occupancies. Binding and unbinding are, however, intrinsically stochastic, and, more generally, noise is ubiquitous in biological systems (*Balázsi et al., 2011*).

In the context of PCP, the noise can arise from a variety of sources with a range of timescales. Some of the sources of noise are cellular processes like protein production and degradation and cell division, growth, birth, and death, which can affect the total protein concentration of Ft and Ds in each cell. These can lead to variations in protein concentrations across the tissue. We have examined the impact of such stochastic effects on the polarization in the tissue.

We have incorporated static or 'quenched' noise in the equations as described earlier (*Equation 15*). In order to retain some connection between the state with quenched disorder and the reference uniform system, we have assumed the noise amplitude (variation in the protein levels across the tissue) to be small compared to the average concentration $\rho$ of protein in each cell. We start with calculating the polarization in the tissue $p$, averaged over multiple realizations, for a range of noise amplitude $S$. We observe that the tissue starts losing polarization if the noise level exceeds a threshold (*Figure 3A*).

The other source of noise in PCP arises from the nature of the protein binding and unbinding from the membrane or the inter-cellular interactions at the cell-cell interface (*Burak and Shraiman, 2009*). We incorporated this dynamical noise in the equations as described earlier (*Equations 10–12*). Again, we have considered the noise amplitude to be not very large as compared to a scale determined by $\beta_0/\gamma$, the characteristic rate of the protein kinetics in this system. We again calculated polarization in the tissue $p$, averaged over multiple realizations, for a range of noise amplitude $\eta$. Here also, we observe a decrease in $p$ as the noise amplitude $\eta$ is higher than a threshold level (*Figure 3—figure supplement 1A*).

This decrease in the average tissue polarization due to both types of stochasticity can be attributed to two mechanisms. It can either be a result of a decrease in the polarity magnitude of each cell, or the polarity magnitude for each cell remains intact, but its direction becomes uncoordinated with noise.

To understand this, we looked at the polarity of each cell $p(i)$ in the tissue for different amplitudes of static and dynamic noise (*Figure 3B–E*, *Figure 3—figure supplement 1B-E*).When the noise amplitudes are small ($S = 10^{-5}$, $\eta = 10^{-4}$), all the cells in the tissue are polarized in the same way.As we increase the noise amplitude ($S = 10^{-4}$ or $\eta = 10^{-3}$), cells in the tissue form patches of polarizations of similar magnitude but of opposite directions. As the noise amplitude increases further, the number of these patches increases, resulting in a loss of coordinated tissue polarity. It must be noted that the cells at the interface of any two patches of opposing polarity also show a decrease in the polarity magnitude to maintain a continuous transition of polarity from one direction to another.

These observations show that over the parameter range studied, the noise predominantly influences the overall coordination of the polarity in the tissue with minimal effect on the PCP of individual cells. These observations for the noise in protein kinetics are in agreement with the results in *Burak and Shraiman, 2009*, where a decrease in tissue polarity is shown if total protein levels are small. As argued in *Burak and Shraiman, 2009*, the effect of such dynamic noise in PCP can be considered to be similar to that of the temperature in classic magnetization models where an increase in temperature results in the loss of magnetization. On the other hand, the static noise is like 'quenched disorder' in condensed-matter systems (*Harris, 1974*; *Radzihovsky, 2015*).

Following the similarities of PCP with other well-studied systems in statistical physics, we close this subsection with a word of caution. As we remarked above, any nonzero noise should destroy order (i.e. spontaneous macroscopic polarity) in the 1D models. So, in the 1D models, what might appear to

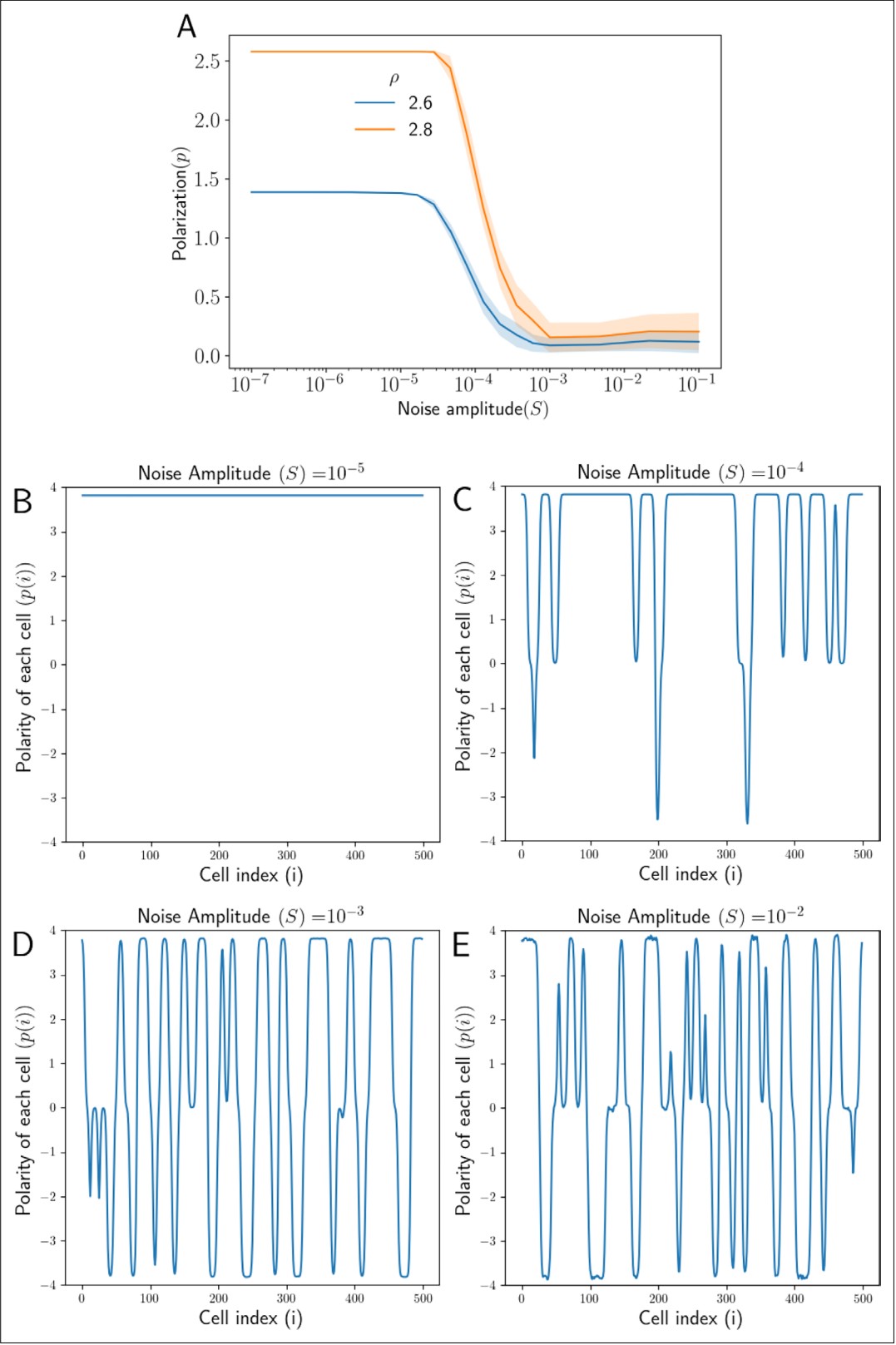

**Figure 3.** Effect of noise in total protein concentrations on polarization. (**A**) Tissue Polarization $p$ as a function of noise $S$ for different values of mean protein levels $\rho$. The shaded regions mark the standard deviations across the 50 simulations. (**B–E**) The polarity of each cell $p(i)$ in the tissue for four representative values of noise $S$. For

*Figure 3 continued on next page*

*Figure 3 continued*

an increase in the noise in the protein levels, cells form patches of opposing polarizations, resulting in the loss of overall tissue polarity.

The online version of this article includes the following figure supplement(s) for figure 3:

**Figure supplement 1.** Effect of noise in protein kinetics on polarization.

be an ordered phase is simply a low-noise state in which the correlation length, while finite, exceeds the system size.

## Swirling PCP patterns are possible in the absence of gradient in the expression

Several experimental works (*Ma et al., 2003*) report swirling patterns in the PCP, especially when the global cue, that is, the tissue level expression gradient, is absent (*Matis and Axelrod, 2013*). We, therefore, explore the occurrence of non-uniform polarization of the tissue within our model. Swirls being a two-dimensional phenomenon, we study the case of a 2D hexagonal lattice, which is a fair approximation to the actual epithelial geometry. We start with the case where the protein expression does not have any tissue-level expression gradient. Similar to the 1D case, we observe that the tissue gets polarized if the total protein concentration is above a threshold value (*Figure 4A*).

Noise (both due to variation in total protein concentration and protein kinetics) in our two-dimensional model leads to non-homogeneous polarization in the tissue (*Figure 4B*). We observe a variety of patterns, some of which have been observed experimentally (*Matis and Axelrod, 2013*). First, we observe aster and swirling patterns where polarization (**P**) in multiple cells either originate or terminate, or rotate around a point, respectively (see representative examples in *Figure 4—figure supplement 1*, respective curl and divergences of the polarization field in *Figure 4—figure supplement 2*, and time evolution in *Figure 4—video 1*, *Figure 4—video 2*). For static noise, these patterns are static in time once the cells are polarized. This interesting finding accounts qualitatively for the swirling patterns observed in experimental systems (*Ambegaonkar and Irvine, 2015*; see *Axelrod, 2020*; *Strutt and Strutt, 2021* for reviews) or at least suggests that they are a natural outcome of PCP physics. These types of swirling patterns have also been studied recently in out-of-equilibrium disordered active systems (*Chardac et al., 2021*). These results also capture the experimental findings

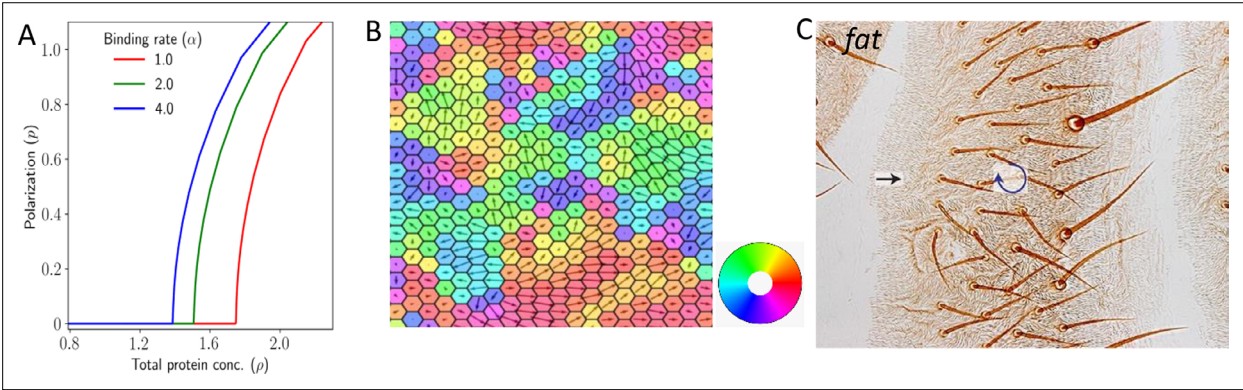

**Figure 4.** Polarization in 2D lattice in the absence of expression gradient. (**A**) Polarization magnitude |**p**| as a function of total protein concentration $\rho$. (**B**) A representative example of tissue polarization in the presence of static noise with $S = 10^{-4}$. (**C**) Experimental image showing swirling patterns in case of Ft mutants in *Drosophila* wing (reproduced from Figure 10A of *Ambegaonkar and Irvine, 2015*).

The online version of this article includes the following video and figure supplement(s) for figure 4:

**Figure supplement 1.** PCP patterns in the presence of static noise.

**Figure supplement 2.** Quantification of asters and swirls in PCP.

**Figure 4—video 1.** Time evolution videos of non-uniform polarization in 2D without gradient.

https://elifesciences.org/articles/84053/figures#fig4video1

**Figure 4—video 2.** Time evolution videos of polarization when additive white noise has been added to the protein kinetics equations.

https://elifesciences.org/articles/84053/figures#fig4video2

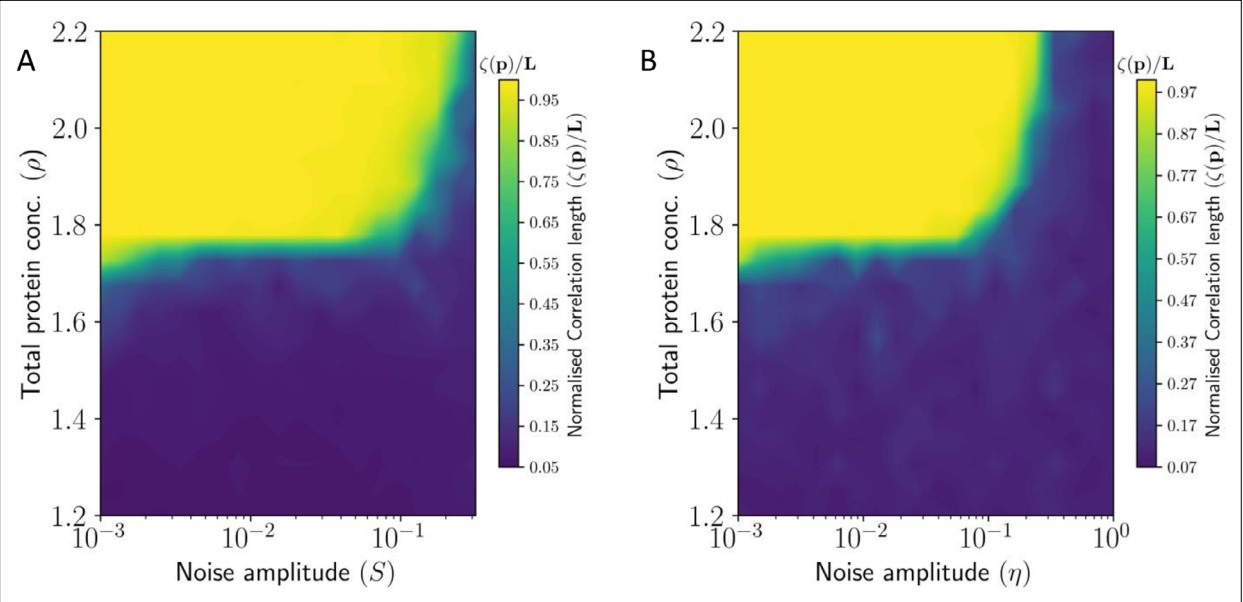

**Figure 5.** Heat Map showing correlation length $\zeta$ of PCP in tissue as a function (**A**) static noise $S$, and (**B**) dynamic noise $\eta$, and total protein concentration $\rho$. Here the correlation length has been normalized by the system size $L$.

observed by *Ambegaonkar and Irvine, 2015*; *Figure 4C*. This shows that in the absence of the expression gradient in 2D, the PCP is coordinated 'locally' but not 'globally'.

To quantify the length scale of 'local' coordination in PCP in the presence of noise, we calculate the correlation length $\zeta$ of the tissue-level polarization vector field. $\zeta$ is the characteristic length over which the tissue demonstrates a coordinated PCP.

The PCP correlation length $\zeta$ for static $S$ and dynamic $\eta$ noise are shown in figures *Figure 5A* and *Figure 5B*, respectively. We observe that for higher protein concentrations (large values of $\rho$) higher magnitude of noise is required for complete loss of coordination in PCP. This means that the system is robust against static and dynamic perturbations if the local interactions are strong enough. Of course, for very high noise levels the high values of $\rho$ are unable to rescue the coordinated tissue polarization.

## Gradients in protein concentrations facilitate PCP and provide global cue

As mentioned earlier, the proteins of the global module of PCP are expressed as a gradient at the tissue level (*Matis and Axelrod, 2013*). These gradients have been studied extensively in the context of wing development in *Drosophila* (*Figure 6A*). Therefore, we also investigated the effect of the expression gradients of the proteins on PCP establishment and its alignment with the tissue axis. For simplicity, we start with studying the effects of a linear gradient of Ft. Gradients of Ds have similar effects. The choice of a linear gradient is inspired by experimental studies (*Matakatsu and Blair, 2004*; *Ma et al., 2003*; *Simon, 2004*) that demonstrated the presence of gradients that are either linear (*Hale et al., 2015*) or can be represented by piece-wise linear approximations. Several other modeling works *Mani et al., 2013*; *Jolly et al., 2014* have also taken gradients to be linear in the position of cells in the tissue.

We incorporate the linear gradient in Ft expression such that the total protein concentration of Ft varies in the tissue as

$$f_T = \rho + \epsilon x_i \tag{39}$$

with $\epsilon \ll \rho$ and $\rho$ is the concentration of Ft at some reference location in the tissue. The reference tissue location mentioned here was selected appropriately to ensure protein levels do not become negative in any cell. Here $\epsilon$ decides the steepness or the strength of the gradient in the tissue and $x_i$ is the position of the cell (having index $i$) in the tissue. The concentration of Ds is taken to be uniform throughout the tissue and is equal to $\rho$. We studied the effect of expression gradients in both 1D and

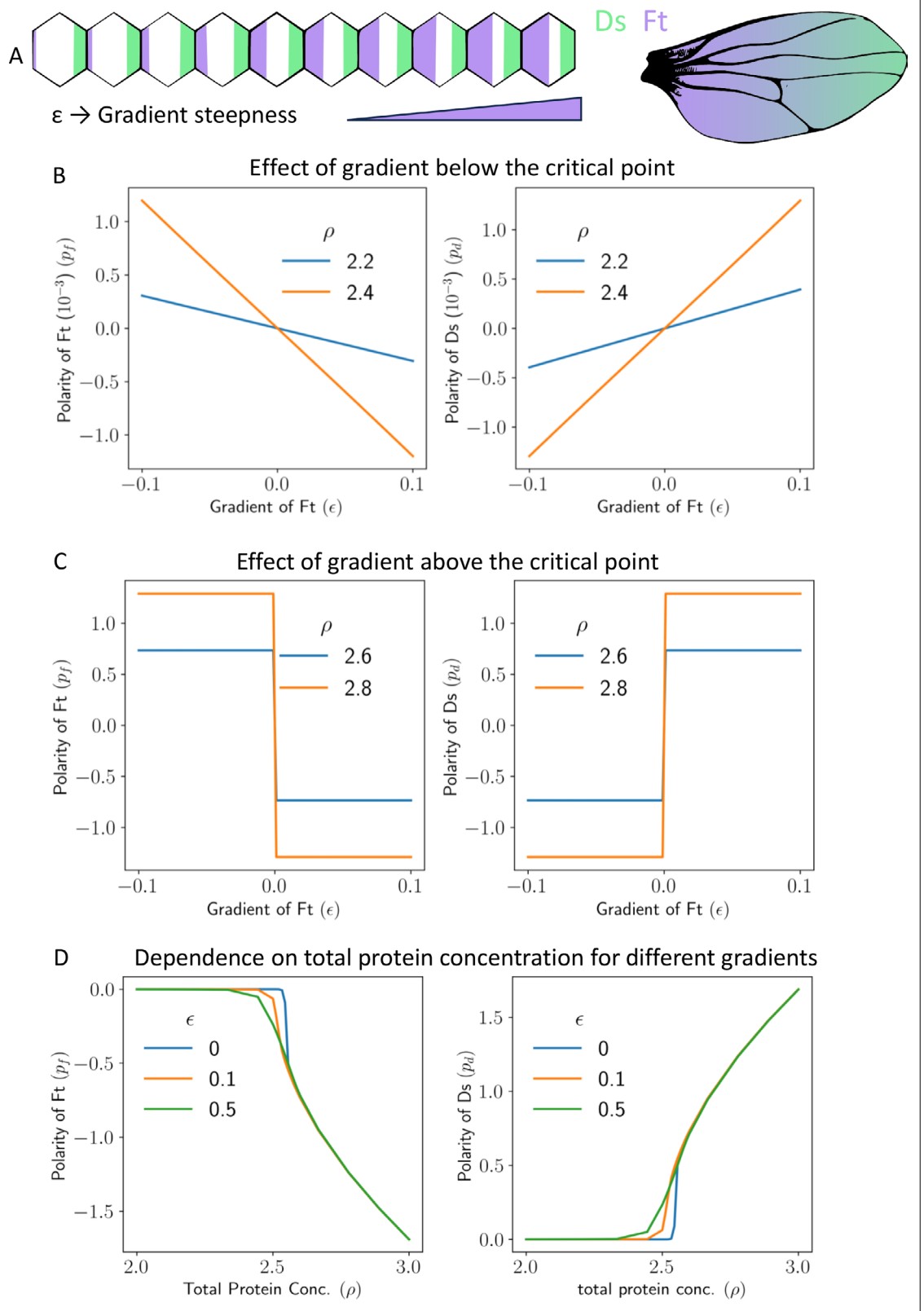

**Figure 6.** Gradient can polarize the tissue and decide its direction. (**A**) Schematics showing expression gradient of Ft in 1D model (left), and an example *Drosophila* wing with gradient expression levels. (**B**) Polarity of Ft ($p_{Ft}$) and Ds ($p_{Ds}$) as a function of gradient of Ft ($\epsilon$), when total protein concentration ($\rho$) is below the critical point. (**C**) Polarity of Ft ($p_{Ft}$) and Ds ($p_{Ds}$) as a function of gradient ($\epsilon$) when total protein concentration ($\rho$) is above the critical point. (**D**) Polarity of Ft ($p_{Ft}$) and Ds ($p_{Ds}$) as a function of total protein concentration ($\rho$) for different values of gradient ($\epsilon$).

*Figure 6 continued on next page*

*Figure 6 continued*

The online version of this article includes the following figure supplement(s) for figure 6:

**Figure supplement 1.** Gradient decides the direction and maintains polarization in 2D.

2D. In both cases, the results are largely similar (*Figure 6* for 1D, and *Figure 6—figure supplement 1* for 2D).

If $\epsilon$ is positive, then, for each cell, the neighbor towards the right has more total concentration of Ft, and the one on the left has a less concentration as shown in *Figure 6A*. Recall that, in the absence of any gradient, the tissue is unpolarized if the total concentrations of Ft and Ds are below their critical values. This, however, is not the case when a gradient of expression is present. In the presence of a gradient, the tissue is polarized even if total protein concentration is below threshold critical ($\rho < \rho_c$; *Figure 6B* and *Figure 6—figure supplement 1A*). The magnitude of polarization, however, is small in this case for weak gradients. In fact, in this case, the magnitude of polarity in Ft and Ds is proportional to the steepness $\epsilon$ of the expression gradients and also on $\rho$.

This effect of the weak gradient on PCP can be understood from the continuum model. We solve *Equations 29–32* with

$$f_T = \rho + \epsilon x \tag{40}$$
$$d_T = \rho \tag{41}$$

for the case where total protein concentration is below the critical protein concentration and $\epsilon \ll \rho$. For this, we get the steady state (see Appendix 1 for details)

$$p = \left( \frac{2\rho_m^2}{\rho_m^2 - 4} \right) \frac{2\alpha\epsilon}{2\alpha + 1}. \tag{42}$$

This equation tells us that the tissue will always be polarized in the presence of the gradient. The slope of polarization ($p$) vs ($\epsilon$) is determined by ($\rho_m$), which in turn is determined by the protein binding rate ($\alpha$) and total protein concentration of Ds ($\rho$).

Interestingly, in the presence of the expression gradient, the polarization of Ft ($p_f$) is always in the direction opposite to that of the gradient. We can understand this observation in the following way. For a given cell, if the neighbor on the right has more Ft concentration than the one on the left, then Ds will localize more towards the right because of the intercellular heterodimer formation. Ft from the neighbor towards the right of each cell will start localizing towards the left side of the neighbor to form a heterodimer with Ds. This feedback loop amplifies their initial differences over time and will result in the eventual localization of Ft towards the left side of all the cells and Ds on the right side.

If the total protein concentration of Ft and Ds is above the threshold protein concentration, the tissue polarizes due to intercellular interactions, but the gradient decides the direction of polarization. Due to the combined effect of gradient and spontaneous polarization, in this scenario, the cells get strongly polarized (compare the values on the $y$ axes of *Figure 6C* and *Figure 6—figure supplement 1B*). The polarization of Ft is again opposite to the gradient direction. The physics of spontaneous polarization implies that an arbitrarily weak gradient decides the direction of polarization, and the tissue then polarizes due to the feedback loop of intercellular interactions (*Figure 6C* and *Figure 6—figure supplement 1B*).

We also looked at the effect of noise in protein kinetics to understand how noise disrupts the directional cue provided by the gradient (*Figure 7A*). We calculated the polarity of Ft ($p_f$) and Ds ($p_d$) as a function of the gradient of Ft ($\epsilon$) for different noise amplitudes ($\eta$). We observe that the gradient still decides the direction polarity in the presence of noise, but for weak gradients, the effect of noise dominates with a reduction in overall polarity in the tissue.

Next, we study how a gradient affects the robustness of polarization in the presence of noise. We calculated the polarity of Ft ($p_f$) and Ds ($p_d$) as a function of noise amplitudes ($\eta$) for different gradient steepness of Ft ($\epsilon$; *Figure 7B*). We observe that the higher the gradient, the higher the noise amplitude needed for the system to lose the polarization. Therefore, the gradient helps in the maintenance of the polarized state, and a system with a higher gradient is more robust to perturbations. We also studied the effect of static spatial noise on polarization (*Figure 7—figure supplement 1A and B*), and the results are largely similar to that of time-dependent noise in protein kinetics (*Figure 7*). Finally, we

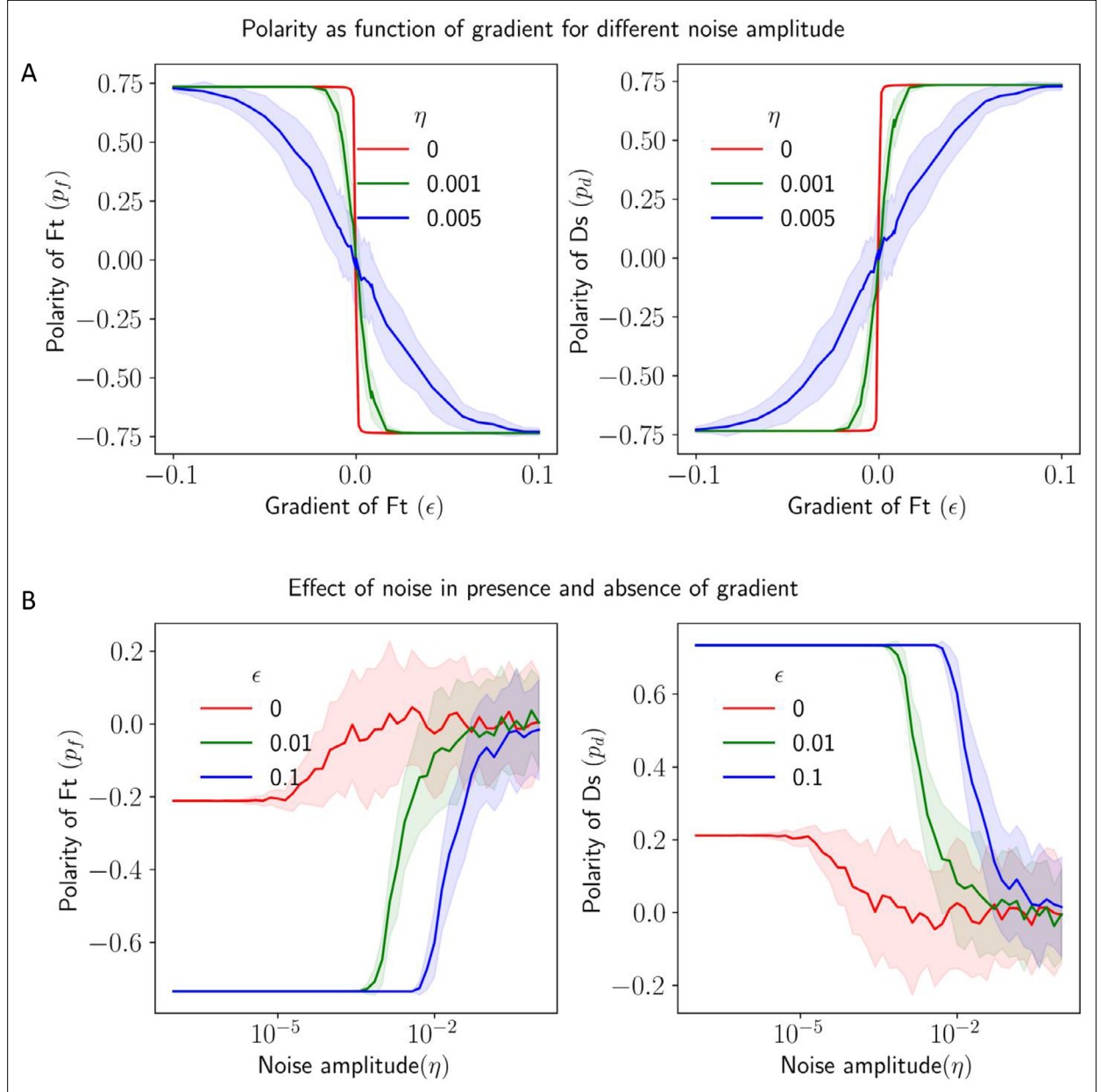

**Figure 7.** Effect of noise in protein kinetics on the polarization in the presence of gradient cue. (**A**) Polarization of Ft ($p_f$) and Ds ($p_d$) as a function of gradient ($\epsilon$) with noise. (**B**) Polarization of Ft ($p_f$) and Ds ($p_d$) as a function of noise amplitude ($\eta$) in the presence of expression gradient. In each panel, the solid lines and the shaded regions mark the average and standard deviations in the polarity over 50 simulations, respectively.

The online version of this article includes the following figure supplement(s) for figure 7:

**Figure supplement 1.** Effect of static spatial noise on the polarization in the presence of gradient cue.

study the effect of total protein concentration ($\rho$) on the polarization of Ft ($p_f$) and Ds ($p_d$) for various gradient amplitudes ($\epsilon$) (**Figure 6D** and **Figure 6—figure supplement 1C**). An increase in the gradient shifts the transition between weak and strong polarization of the cells to smaller values of $\rho$.

These results show that expression gradients of the proteins in the PCP also play other roles in addition to providing global cue to align PCP direction with the tissue axis. A gradient not only helps in polarity establishment when overall protein levels are low but also stabilizes the polarized state as well.

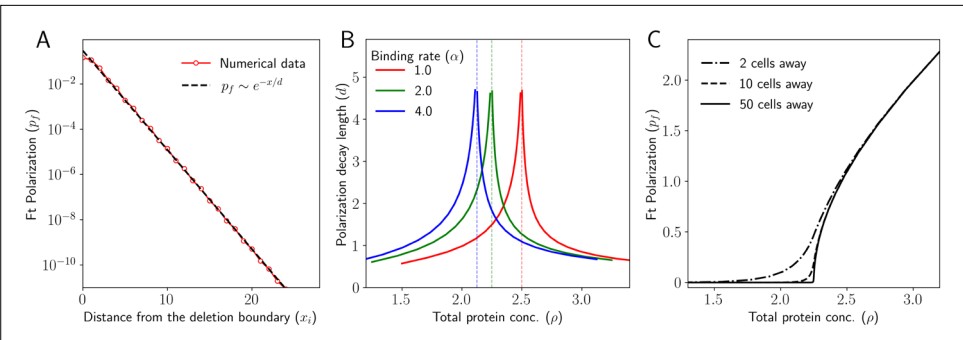

**Figure 8.** Deletion induces a polarity that decays exponentially. (**A**) Ft polarization ($p_f$) of each cell as a function of distance from the deletion boundary. (**B**) Decay length of polarization as a function of total protein concentration. (**C**) Polarization at different locations in the tissue relative to the deletion boundary.

## Down-regulation of Ft from a region in the absence of gradient leads to exponentially decaying isotropic polarity defects

In experimental investigations, in order to identify the role of the protein interactions and their non-cell autonomous effects, the protein expression levels are perturbed from select regions of the tissue (*Ma et al., 2003*).

In one such experimental strategy, Ft or Ds are down-regulated or up-regulated from a selected tissue region, and its fallout is observed in the cells of that region and outside of it. This scenario has been studied qualitatively in other modeling works (*Burak and Shraiman, 2009*; *Jolly et al., 2014*; *Hale et al., 2015*) as well. In the following, we are going to look at it quantitatively.

In order to recapitulate this scenario in silico, we set the total concentration of Ft in a region of the 1D tissue (the left half) to be zero while keeping it uniform and nonzero in the right half of the tissue. On the other hand, the total protein concentration of Ds is kept uniform and non-zero.

We observe that deleting Ft from a region induces a polarization that decays as we move away from the deletion boundary (*Figure 8A*). Ft from the cell towards the right of the deletion boundary binds to the Ds from the cell towards the left of the deletion boundary. This leads to more Ft localizing towards the left and Ds towards the right. As we move away from the deletion boundary, the effect of these inter-cellular interactions decays. We fitted the decay of polarity to a decaying exponential function and calculated the decay length of the polarization, i.e., the length or number of cells after which the polarization reduces to $e^{-1}$ of its value at the deletion boundary. Next, we calculate this decay length as a function of total protein concentration (*Figure 8B*). Finally, we plot the polarization as a function of total protein concentration at several distances from the deletion boundary. Specifically, we calculate the polarization as a function of total protein concentration at the deletion boundary and at points that are 2, 10, 50 cells away from the deletion boundary and for the case when there is no deletion in the tissue (all cells have equal Ft and Ds). As expected, as you move away from the deletion boundary (50 cells away), the tissue behaves like the case when there is no deletion at all. (*Figure 8C*). To understand why polarity is formed at the deletion boundary and why it follows an exponential decay profile, we use our continuum model, which can be solved analytically (see Appendix for details). We get

$$\mathrm{p} = \begin{cases} 0, & \text{if } \rho_m \leq 2 \\ p_0 e^{-x/d}, & \text{if } \rho_m > 2. \end{cases} \tag{43}$$

where $d = \dfrac{\rho_m^2}{\rho_m^2 + 4}[\alpha(\rho_m^2 - 4)/8]^{-1/2}$ is the decay length of polarity and $p_0$ can be determined by the boundary condition at $x = 0$. The continuum model also predicts the exponential decay in polarity, which is in agreement with our discrete model results.

Next, we deleted or up-regulated Ft from a circular region of the tissue in the 2D model. The total protein concentration of Ft is the same as that of Ds throughout the tissue but in the circular region

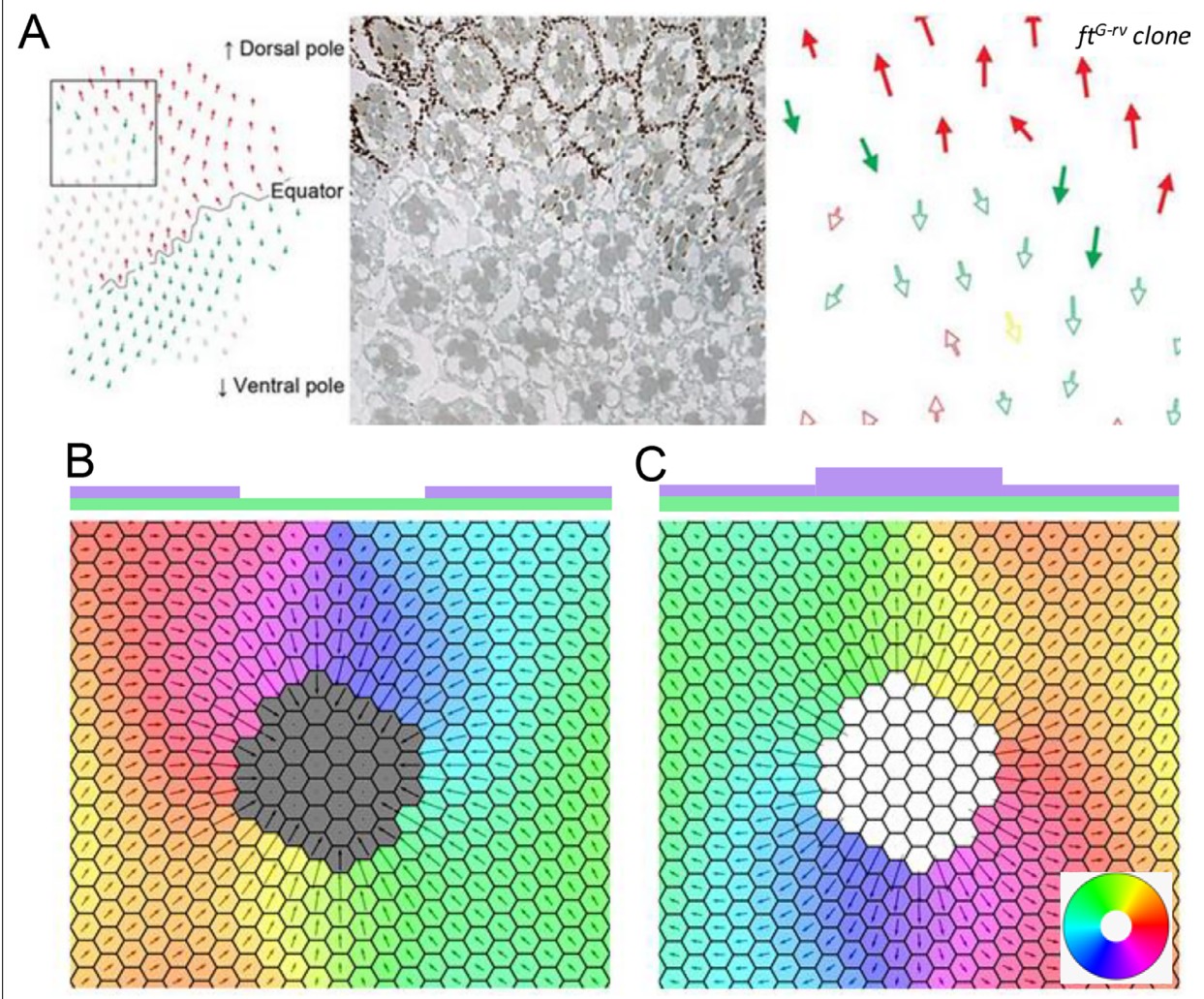

**Figure 9.** Deletion and upregulation in a 2D layer of tissue. (**A**) Experimental images showing a reversal in polarity in *Drosophila* ommatidia in ft mutant clones (reproduced from Figure 2A of *Sharma and McNeill, 2013*). (**B**) Polarization when Ft is removed from a region of the tissue. (**C**) Polarization when Ft is up-regulated in a region of the tissue. The polygon colors in B and C denote the direction of polarization in each cell. Time evolution of polarization when one of the genes is deleted or upregulated from the circular region at the center of the tissue. The system gets polarized radially from the deletion region.

© 2013, The Company of Biologists Ltd. Panel A was reprinted from Figure 2 A of *Sharma and McNeill, 2013*, with permission from The Company of Biologists Ltd.; permission conveyed through Copyright Clearance Center, Inc. It is not covered by the CC-BY 4.0 license and further reproduction of this panel would need permission from the copyright holder.

The online version of this article includes the following video for figure 9:

**Figure 9—video 1.** Time evolution of polarization when one of the genes is deleted or upregulated from the circular region at the center of the tissue.
https://elifesciences.org/articles/84053/figures#fig9video1

where it is either zero or a fixed high value. Our aim is to account for the experimental observation that cells near the deletion boundary are polarized towards or away from the deletion region if a gene is deleted or up-regulated in that part, respectively (*Figure 9A*, image taken from *Sharma and McNeill, 2013*).

When Ft is deleted from a region, and the total protein concentration of Ft and Ds is above the threshold value, most of the cells in the tissue polarize radially in the direction of the deletion region (*Figure 9B*; *Figure 9—video 1*).

When Ft is up-regulated in the region, the results are the opposite of the deletion case. Now, cells are polarized away from the upregulation region (*Figure 9C*).

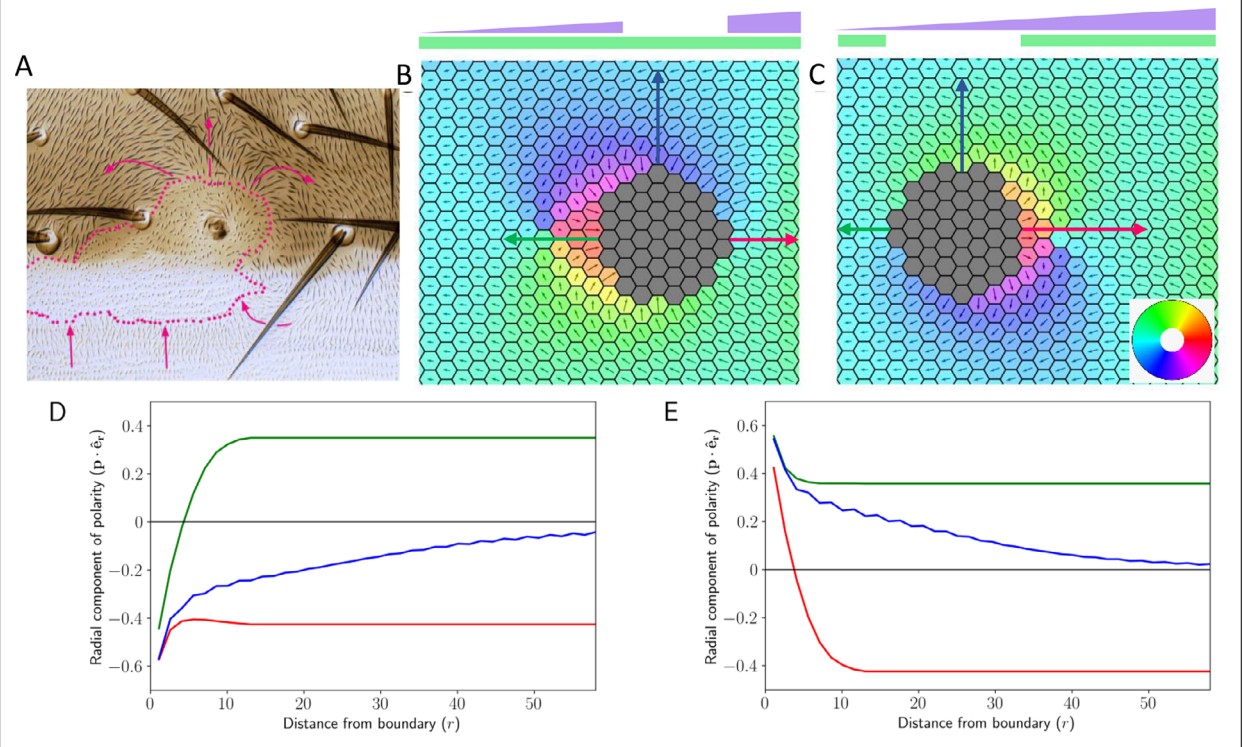

**Figure 10.** Deletion and gradient can explain domineering non-autonomy. (**A**) Experimental image showing the effect of localized perturbation in pre-hairs on *Drosophila* wing (reproduced from Figure 6A of *Chorro et al., 2022*). (**B**) Cell polarization in the tissue in the presence of Ft gradient and its localized loss. (**C**) Cell polarization in the tissue when Ds has been removed locally while Ft is expressed in a gradient manner. The colors in B and C denote the direction of cell polarization. (**D** and **E**) Polarity in the radial direction as a function of distance from the deletion boundary. The color of the trajectory corresponds to the direction in which the arrows point in B (**D**) and C (**E**). Time evolution of polarization when one of the proteins is deleted from the central region of the tissue along with a gradient of Ft throughout the tissue.

The online version of this article includes the following video for figure 10:

**Figure 10—video 1.** Time evolution of polarization when one of the proteins is deleted from the central region of the tissue along with a gradient of Ft throughout the tissue.

https://elifesciences.org/articles/84053/figures#fig10video1

We can understand these results by simply thinking of the feedback loop. If Ft is deleted from a region, the Ft from the surrounding region will make a heterodimer with the Ds of the cells inside the deletion boundary. This will, in turn, result in the localization of Ds from the surrounding region moving away from the deletion region, which results in Ft from the next circular layer of cells localizing inwards. This feedback loop will result in polarization of the tissue in the direction of deletion. A similar process happens in the case when Ft is up-regulated, except that the directions will be reversed.

## The presence of gradient and local down-regulation produces domineering non-autonomous effects

Finally, we tried to understand one of the most interesting features of PCP- domineering non-autonomy (see *Figure 10A*, image taken from *Chorro et al., 2022*). In domineering non-autonomy, disruption in protein levels from a region of tissue affects the polarity of neighboring wild-type cells (*Amonlirdviman et al., 2005*; *Axelrod and Tomlin, 2011*; *Ambegaonkar and Irvine, 2015*). For this, we incorporated the gradient in the expression levels of Ft, and Ds expression was kept uniform in the tissue. We looked at the two cases- first, where Ft was removed from a region of tissue (*Figure 10B*), and second, where Ds was removed from a region of the tissue (*Figure 10C*).

In both of these cases, we observed that the polarization of the cells is dependent on their relative position (in terms of both distance and orientation) to the deletion region. Expectedly, the cells far from the deletion region follow the polarity, which is primarily dictated by the expression gradient of Ft. However, close to the deletion region, the prominent effect is seen on different sides of the

deletion region for Ft and Ds deletions (compare *Figure 10B* and *Figure 10C*). Since in the absence of tissue level gradient, the tissue polarity in the neighborhood of a clone is radial (*Figure 9*), we quantified the radial component of the polarization in the presence of tissue level gradient of Ft (*Figure 10D–E*). In the direction normal to that of Ft gradient, the radial polarity component decays to zero with distance (see blue curves). In the direction parallel to the that of Ft gradient, however, the far-field polarity is still driven by the Ft gradient (see a magnitude of $|p| \approx 0.4$). In the cells close to the clone, however, the polarity magnitude is determined by the distance and relative location of the cell from the clone (see green and red curves). This behavior can be explained by the superposition of the effects of Ft gradient (*Figure 6*) and loss of Ft or Ds (*Figure 9C*) from the deletion region. The gradient polarizes the whole tissue uniformly, whereas deletion of Ft or Ds causes radial polarization of tissue in the absence of Ft gradient. When both of these are present in the tissue, the combined effect is reflected in the disruption of polarity on the two sides of the deletion region (*Figure 10—video 1*).

Even though the experimental evidence for such non-autonomous effects in PCP due to a local loss of the proteins of 'core' modules of PCP is now undisputed (*Strutt and Strutt, 2021*), the same for perturbations only in Ft and Ds expressions still is not yet settled. As reported in *Strutt and Strutt, 2002*, the loss of Ft from the localized region in *Drosophila* wing results in strong effects in PCP, showing non-autonomous swirling patterns in downstream indicators of PCP. Similarly, (*Ma et al., 2003*) has also reported the effect of localized Ft loss on prehairs in *Drosophila* wings. A systematic investigation is, therefore, required to quantify the asymmetric localization of global proteins in the presence of localized Ft or Ds loss. Such a study can take help from the quantified predictions of the model to test for the non-autonomous effects of localized Ft and Ds perturbations.

## Discussion

The minimal discrete model presented here offers a useful framework for combining the role of global (expression gradients) and local (heterodimer formation) interactions in PCP and studying their collective effects. This framework builds upon the discrete modeling of global/local modules (*Amonlirdviman et al., 2005*; *Hale et al., 2015*) and other phenomenological models (*Hazelwood and Hancock, 2013*; *Mani et al., 2013*).

Since experimental studies haven't confirmed the interaction of heterodimers per se, the current model based on protein-protein interactions (as opposed to heterodimer-heterodimer interactions *Mani et al., 2013*; *Shadkhoo and Mani, 2019* which can be considered an outcome of protein-protein interactions) provides a better-motivated framework to understand PCP. Some earlier studies like *Hazelwood and Hancock, 2013*; *Burak and Shraiman, 2009* have invoked the similarities between the XY model of ferromagnetism and PCP formation. This has helped in understanding the formation of swirling patterns and the effect of a gradient in the tissue level polarization.

Due to its minimalist nature, the model presented here also differs from other protein-protein interactions based studies, such as (*Amonlirdviman et al., 2005*; *Hale et al., 2015*; *Fisher and Strutt, 2019*), which involve a high number of model parameters, and therefore difficult to analyze (*Gutenkunst et al., 2007*).

In this work, we could also analyze the discrete model in the continuum limit, which gives some valuable insights that are validated systematically over a wide range of parameter values in the discrete version. After non-dimensionalizing, we are left with just four free parameters: protein binding rate ($\alpha$), the total protein concentration of Ft and Ds ($\rho$), and gradient steepness ($\epsilon$), and $k$, the Hill's coefficient. Among these parameters, $k$ has no qualitative influence on PCP. Therefore, we are left with three free parameters that significantly impact PCP and are not too difficult to control and study. Conceptually, the first two parameters ($\alpha$ and $\rho$) correspond to the 'strength' of the relay of PCP 'signal' from one cell to another, while the last one ($\epsilon$) modulates the direction of signal propagation and offers robustness against any stochastic perturbations. In the absence of a gradient, the polarity is still established in individual cells but is not necessarily coordinated at the tissue level. When the local interaction is weak (total protein concentration is low), the global interaction is responsible for the formation and direction of polarity. On the other hand, when local interaction is strong, the gradient helps in providing the directional cue and maintenance of polarity in the presence of perturbation.

Thus, the minimal model can explain diverse experimental observations seen in many biological systems, suggesting common design principles at play.

Even though the results in this work are presented for the proteins of the global PCP module, similar arguments can also be extended to the core module proteins as well. Some of the PCP features demonstrated by the model with perturbed Ft and Ds, such as the formation of swirls, polarity reversal in the presence of local loss, and domineering non-autonomy, have also been reported when proteins from the core module are perturbed (*Humphries et al., 2020*; *Strutt et al., 2023*; *Cetera et al., 2023*). In fact, a recent experimental work exploring sequential event of PCP in core proteins has shown stabilization of Fmi on the cell membrane due to the presence of Fmi-Fz complex on the neighboring cell (*Strutt et al., 2023*), which is then followed by the recruitment of Stbm. It has to be noted that the model proposed here only includes intercellular interactions. In the core pathway, however, the intracellular interactions (by Pk, Dsh etc.) have also been known to play a crucial role in the stabilization of the membrane protein complexes. Therefore, such intra-cellular interactions must also be considered before the model can be applied to the core pathway.

The quantitative nature of the analysis in this work has resulted in predictions which can be tested experimentally.

1. Effect of Ft and Ds interactions and transport: The effect of binding affinity is reflected in the critical protein concentrations required for the polarity establishment (*Figure 2C*). The critical levels (in dimensional form) of Ft and Ds for PCP establishment are $\rho_c = \dfrac{1}{\gamma}\left(\rho_{c0} + \dfrac{1}{2\alpha}\right)$ where $\rho_{c0}$ depends on the model dimensions. Here, $\gamma$ depends on specific structural forms of Ft and Ds, and their biochemical modifications with large value of $\gamma$ for stronger binding affinity. On the other hand, $\alpha$ is regulated by the mechanisms of intracellular protein transport which is independent of minor structural modification in Ft and Ds. A recent work has identified amino acid modifications at the Ft-Ds extracellular domain using biochemical assays which result in reduced binding affinity (*Medina et al., 2023*). These targeted modifications in the sequences of Ft and Ds, along with biochemical assays, can establish a quantitative dependence between $\gamma$ and $\rho_c$. In fact, this approach can also be utilized to compare the present model assumptions with those based on intracellular heterodimer interactions (*Mani et al., 2013*; *Burak and Shraiman, 2009*; *Fisher and Strutt, 2019*). The role of *cis* inhibition between the heterodimers of opposing polarity at the cell-cell interface has been proposed to be important for PCP and gradient sensing. The nature of this *cis* inhibition can be modulated with targeted amino acid modifications in the Ft and Ds in their cytoplasmic domains. Another experimental approach of interest is in vitro interaction of cells expressing only Ft or Ds (*Loza et al., 2017*) to study the protein dynamics at the cell-cell interface. This work shows that in the co-culture of the cells expressing only Ft and only Ds, the cell-cell interface demonstrates slower dynamics of Ft and Ds, implying stability of Ft-Ds complex without any intracellular *cis* interaction. This experimental approach can further be extended to in vitro investigation with forms of Ft and Ds which have been modified at their cytoplasmic domains to further present evidence of any *cis* inhibitory mechanisms.

2. Correlation length of PCP: Correlation length is one of the most important parameters for the quantification of order in physical systems. Still, this has not been quantified and explored in most modeling or experimental works, barring a few such as (*Burak and Shraiman, 2009*; *Mani et al., 2013*). We have quantified its dependence on the total protein levels and the stochastic effects (*Figure 5*). Experimental estimation of the correlation lengths in PCP for wildtype and for genetic perturbations, such as uniform overexpression and/or loss of Ft and Ds, is yet to be done.

3. Effect of tissue-level protein expression gradients: The results shown here point toward the dual role of tissue-level protein gradients in PCP. This is in contrast to the models which consider gradient to be only a directional guide for polarity coordination and not polarity establishment. As shown in *Figure 5B* and *Figure 6A* an increase in the strength of tissue level protein gradient results in an increase in tissue polarity. In fact, in the presence of a tissue-level gradient, a higher magnitude of stochasticity is required to disrupt the tissue scale PCP (*Figure 6B*). The role of expression gradient providing robustness to the PCP also requires experimental investigation. One possible way to realize this can be by modulating the stochasticity in the system either by a variation of temperature (dynamic noise) or by loss/gain of Ft and/or Ds in several relatively small regions of the tissue (static noise) while maintaining the tissue level expression gradients.

4. PCP decay length from a boundary: As shown in *Figure 7*, the polarity decays exponentially with distance from a boundary. The non-autonomous effect of localized Ft loss has been reported earlier (*Strutt and Strutt, 2002*) but the evidence presented there is qualitative in nature. In

other works involving localized Ft or Ds loss, such as (*Ma et al., 2003*), its effct was observed in terms of the prehairs orientation and location on *Drosophila* wing. The quantifications of such effects (as shown in *Figure 7* and *Figure 10*) due to localized perturbations are yet to be experimentally realized, especially in terms of the asymmetric distribution of Ft and Ds protein on the cell membrane.

It is not that the localized loss or upregulation of proteins – which may not always be very straighforward to experimentally realize – is the only way to generate boundaries. One can also consider existing boundaries in the large epithelial tissues, such as *Drosophila* wing, where veins act like boundaries, albeit not in terms of the protein expression levels but surely in terms of the cell geometry. By modulation of total protein levels along with loss/gain from the tissue the exponential nature of the decay can also be investigated.

5. Domineering non-autonomy: Another experimentally testable prediction from the model is the effect of the protein expression gradients on the domineering non-autonomy, where the loss of the two proteins in the presence of an expression gradient can affect the two sides of the clone differently. On the other hand, in the absence of a gradient, we observe a disruption in polarity in areas adjacent to the mutant clone, but the disruption is isotropic in the sense that the polarization direction near the clone boundary is perpendicular to it. This prediction can also be tested experimentally by perturbing gradients and generating mosaic patterns of deletion or overexpression of specific proteins.

In this model, we have only considered the most simplistic and common interactions seen in the proteins involved in PCP. To keep the model simple, we have only considered the interactions between Ft and Ds, ignoring other proteins of the global module. For instance, Fj, a Golgi kinase that modulates mutual affinities between Ft and Ds by their phosphorylation, and Dachs, an atypical myosin that influences the mechanics of the epithelial cells have not been considered here. In our model, we have considered the small additive noise in protein kinetics and total protein concentration. The underlying assumption here is that the average copy number of proteins is much larger than the levels of noise. But for the tissues where this assumption is not valid, the model needs to be modified (*Shadkhoo and Mani, 2019*).

One of the major limitations of the current model is that it does not account for the change in lattice structure due to cell death, birth, and growth and epithelial flows (*Aigouy et al., 2010*). Modeling studies like (*Shadkhoo and Mani, 2019*) discuss the effect of cell shape changes on polarity formation. Future work needs to incorporate these factors in our discrete modeling framework.

## Acknowledgements

We would like to acknowledge Atchuta Srinivas Duddu and Dhruba Jyoti Mech for the schematics. DS is supported by the KVPY fellowship awarded by the Department of Science and Technology (DST), Government of India. MKJ is supported by the Ramanujan Fellowship (SB/S2/RJN 049/2018) awarded by the Science and Engineering Research Board (SERB), DST, Government of India. SR acknowledges support from a J C Bose Fellowship of the Science and Engineering Research Board, India. MSR would like to acknowledge the financial support provided by the seed grant from IIT Hyderabad and the Startup Research Grant from the Science and Engineering Research Board (SERB), DST, Government of India.

## Additional information

#### Competing interests
Divyoj Singh: The other authors declare that no competing interests exist.

## Funding

| Funder | Grant reference number | Author |
| --- | --- | --- |
| Science and Engineering Research Board | SB/S2/RJN-049/2018 | Mohit Kumar Jolly |
| Science and Engineering Research Board | SRG/2021/001020 | Mohd Suhail Rizvi |

The funders had no role in study design, data collection and interpretation, or the decision to submit the work for publication.

## Author contributions

Divyoj Singh, Formal analysis, Writing – original draft; Sriram Ramaswamy, Supervision, Writing – review and editing; Mohit Kumar Jolly, Mohd Suhail Rizvi, Conceptualization, Supervision, Writing – review and editing

## Author ORCIDs

Divyoj Singh (ID) http://orcid.org/0000-0003-1552-2701
Sriram Ramaswamy (ID) https://orcid.org/0000-0001-7726-8556
Mohit Kumar Jolly (ID) https://orcid.org/0000-0002-6631-2109
Mohd Suhail Rizvi (ID) https://orcid.org/0000-0002-4130-4671

## Decision letter and Author response

Decision letter https://doi.org/10.7554/eLife.84053.sa1
Author response https://doi.org/10.7554/eLife.84053.sa2

# Additional files

## Supplementary files

• MDAR checklist

## Data availability

The current manuscript is a computational study, so no data have bene generated for this manuscript. Modeling code used is available on GitHub page of DS (copy archived at *Singh, 2024*).

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

## Appendix 1

### Analysis of the continuum 1D model of PCP

Uniform protein expressions

For the uniform expressions of proteins and in the tissue, we have (a known constant value). If we assume the total membrane bound Ft and Ds to be equal to each other, that is $f_m = d_m = \rho_m$ (an unknown), we obtain for $k = 2$, the homogeneous solution described by

$$\dot{\rho}_m \approx 2\alpha(\rho - \rho_m) - \frac{4\rho_m}{\rho_m^2 + 4} \tag{44}$$

$$\dot{q} \approx -4q\left(\frac{3\rho_m^2 + 4}{(\rho_m^2 + 4)^2}\right) + \frac{q}{(\rho_m^2 + 4)^4}\left[A_q(\rho_m)p^2 + B_q(\rho_m)q^2\right] \tag{45}$$

$$\dot{p} \approx -4p\left(\frac{4 - \rho_m^2}{(\rho_m^2 + 4)^2}\right) + \frac{p}{(\rho_m^2 + 4)^4}\left[A_p(\rho_m)p^2 + B_p(\rho_m)q^2\right] \tag{46}$$

where $p = p_f - p_d$ and $q = p_f + p_d$, and

$$A_q(\rho_m) = \rho_m^4 - 24\rho_m^2 + 16 \tag{47}$$

$$B_q(\rho_m) = 15\rho_m^4 - 40\rho_m^2 - 16 \tag{48}$$

$$A_p(\rho_m) = 9\rho_m^4 - 8\rho_m^2 + 16 \tag{49}$$

$$B_p(\rho_m) = -7\rho_m^4 - 56\rho_m^2 - 16. \tag{50}$$

We have obtained these relations for a homogeneous state by setting all spatial derivatives to zero in *Equations 29–32*. These equations show that for very small values of $\rho_m$ the only homogeneous solution (where all spatial derivatives vanish) is $q = 0$ and $p = 0$ at the steady state (unpolarized state). For $\rho_m > 2$, however, we get $q = 0$ and $p > 0$ (polarized state) which imply polarization of two proteins in opposite directions. For very large values of $\rho_m$ the system can also have $q > 0$ which implies unequal polarization of the two proteins in the cells. In this work, we do not consider such high levels of $\rho_m$.

Using the equation for $\rho_m$ above we also obtain the critical value of the total protein levels for the onset of polarization to be

$$\rho = 2 + \frac{1}{2\alpha} := \rho_c. \tag{51}$$

This shows that for higher binding rates of the proteins to the membrane, cells can polarize at lower protein expression levels. Please note that we have derived these expressions for $k = 2$ but can also be obtained for any other value of $k$.

### Non-uniform protein expressions

In order to study the effect of a gradient, we consider a weak linear (*Lawrence et al., 2008*) gradient of the expression of Ft such that $f_T = \rho + \epsilon x$ (with $\epsilon \ll \rho$). We assume the following ansatz $f_m = \rho_m + x\tilde{f}_m$, $d_m = \rho_m + x\tilde{d}_m$ with homogeneous polarization levels in the tissue. For $\rho_m \ll 1$, we get from *Equations 29–32*.

$$\dot{p} \approx -4p\left(\frac{4 - \rho_m^2}{(\rho_m^2 + 4)^2}\right) + \frac{8\rho_m^2}{(\rho_m^2 + 4)^2}(\tilde{d}_m - \tilde{f}_m) \tag{52}$$

where

$$\tilde{f}_m - \tilde{d}_m \approx \frac{2\alpha\epsilon}{2\alpha + 1}. \tag{53}$$

In *Equation 52* above we have written only the leading order terms on the right-hand side. For the steady state, we obtain

$$p = \frac{2\rho_m^2}{4 - \rho_m^2}(\tilde{d}_m - \tilde{f}_m).$$

(54)

This shows that in the case of a tissue-level gradient expression of one of the two proteins, the tissue is always polarized and that too in a particular direction which depends on the direction of the gradient. It has to be noted that the expression for the polarization magnitude, as shown above, is valid only for $\rho_m \ll 1$, and higher order terms have to be included in *Equation 52* otherwise.

## Loss of one protein from a region of the tissue

In order to model the loss of Ft from a region of tissue we consider Ft to be absent in the cells at $x < 0$ and focus on the polarization in the cells at $x \geq 0$. This means we are effectively considering a semi-infinite 1D region of $x \geq 0$. The loss of Ft in the region $x < 0$ will set the boundary conditions for the unknown variables $f_m$, $d_m$, $p_f$ and $p_d$ at $x = 0$. In this case, we expect $f_m$, $d_m$, $p_f$ and $p_d$ to be $x$-dependent and their derivatives with respect to $x$ cannot be ignored. Still, we should get the wildtype behavior as seen above for $x \gg 1$. We presume solutions of type $f_m(x) = \rho_m + \tilde{f}_m(x)$, $d_m(x) = \rho_m + \tilde{d}_m(x)$, $p_f(x) = p + \tilde{p}_f(x)$ and $p_d(x) = -p + \tilde{p}_d(x)$ with $|p| \ll \rho_m$. Substituting these values in the *Equations 29; 30* and looking for the steady state solutions gives us

$$\tilde{p}'' = -\frac{\alpha(\rho_m^2 + 4)^2}{8\rho_m^4}(4 - \rho_m^2)\tilde{p}$$

(55)

where $\tilde{p} = \tilde{p}_f - \tilde{p}_d$ is the excess cell polarization due to the loss of Ft in the left half of 1D tissue. Here we have considered only the leading order terms on the right hand side. Depending on the value of $\rho_m$ this equation can have either exponentially decaying solution (for $\rho_m > 2$) or spatially oscillatory solutions (for $\rho_m < 2$). In order to have a 'wildtype' behavior far from $x = 0$ boundary, that is $\tilde{p} \to 0$ as $x \to \infty$, the spatially oscillatory solution becomes unfeasible. This gives us the excess polarization to be

$$(x) = \begin{cases} 0, & \text{if } \rho_m \leq 2 \\ \tilde{p}_0 e^{-x/d}, & \text{if } \rho_m > 2. \end{cases}$$

(56)

where $d = \frac{\rho_m^2}{\rho_m^2 + 4}[\alpha(\rho_m^2 - 4)/8]^{-1/2}$, and $p_0$ can be determined by the boundary condition at $x = 0$. This demonstrates an exponential decay in the magnitude of cell polarization from the boundary of the region where one of the two proteins is not expressed.

