## [Editor Report]

This study presents a valuable model for the emergence of planar cell polarity from the interplay of local interactions and global gradient. The framework of this model is solid. A quality of this model is its simplicity and its convenience for experimental testing.

---

## [Decision Letter]

**Decision letter after peer review:**

Thank you for submitting your article "Emergence of planar cell polarity from the interplay of local interactions and global gradients" for consideration by *eLife*. Your article has been reviewed by 3 peer reviewers, one of whom is a member of our Board of Reviewing Editors, and the evaluation has been overseen by Aleksandra Walczak as the Senior Editor. The reviewers have opted to remain anonymous.

Essential Revisions:

(1) Several models with a similar hypothesis and similar predictions have been published previously. Clearly point out in which way your mechanism fundamentally differs from these previous works. Also discuss, in which way your mechanism is better than those studied in previous works. In particular, explain features that are not explained by other models, and/or generate non-trivial predictions that can be tested experimentally.

In the discussion, you suggest two predictions: One is to measure the decay length around a clone. This is not really a prediction. The other is to look at the asymmetry around a Fat4 clone in the presence of a gradient. This can be interesting, but it was not analyzed in the manuscript, nor was there a serious effort to check if this was observed experimentally.

(2) Some conclusions of this work rely on visual appreciation rather than quantification. This is particularly true for what concerns 2D patterns. An argument of the authors is for example that their model reproduces a variety of known spatial patterns, but the comparison with experiments is only visual. Provide a more quantitative comparison.

(3) Justify better some assumptions underlying the model. Notably, why should Fat-DS complexes be stabilized when bound, and why is additive noise appropriate to account for the effect of stochasticity in the attachment-detachment dynamics?

*Reviewer #1 (Recommendations for the authors):*

More specifically I have the following major comments:

(1) The main claim of the authors is that the new model is 'better' than previous models since it is a relatively simple phenomenological model, yet it focuses on the relevant boundary proteins (and not complexes). While I appreciate the elegance of the new model, it is unclear in what sense is it better. I think the important measure for this question should be its ability to explain experimental behaviors that were not explained before, or generate non-trivial predictions that can be put to experimental test. In the discussion, the authors suggest two predictions: One is to measure the decay length around a clone. This is not really a prediction. The other is to look at the asymmetry around a Fat4 clone in the presence of a gradient. This can be interesting, but this was not analyzed in the manuscript, nor was there a serious effort to check if this was observed experimentally. In summary, the model should not only be elegant but should have non-trivial consequences.

(2) I find it interesting that the assumption that Fat-Ds complexes are stabilized is sufficient to promote global polarity. Most models to date, rely on feedback where complexes in one direction promote the formation of complexes in the same direction and prevent complexes in the opposite direction (Mani et al). The stabilization hypothesis effectively promotes complexes in the same direction (since more Fat and Ds remain bound). However, it is unclear why such interactions can promote complexes in both directions in parallel. I suspect it has something to do with the finite pools of Fat and Ds, but it would be nice to get an intuitive explanation for that.

(3) Just to point out that there are papers that do show enhanced stability of Fat-Ds complexes, and it would be nice to mention these: Hale et al., *eLife* 2015, Loza et al., *eLife* 2017.

(4) When providing images from published work – please provide mutant details (for example in the PCP mutants in Figure 1).

(5) Figure 2 – please provide a plot showing one realization of the simulation showing the local polarity in each cell. It would help clarify the behavior of the local polarity. In 2F, the axis uses peculiar tick numbers. Unclear if the axis is linear or logarithmic.

(6) Section on noise – Please compare to Burak et al. Are the results different/similar?

(7) Figure 3B-E – the x-axis describes tissue position. Please clarify what is the scale of one cell.

(8) Figure 4A – The color scheme is unclear. Specify what is blue and what is red, as well as what the dimensions of each domain (red, white, blue) within the cells refer to.

(9) In Figure 4C it seems that some domains are very large and others are rather small. This is rather different than the swirling observed in some PCP mutants. Can the authors discuss this discrepancy (does it depend on parameters?)?

(10) Figure 4D. The author refers to a manuscript by Michael Simon, but this picture does not appear to be from that paper. As far as I can tell from the image, this is an overexpression of Fat that leads to swirling. Can the authors check if their model captures this behavior?

(11) The authors refer several times to another submitted manuscript (ref 38). One cannot provide results based on unpublished work. Either provide a reference or provide the full analysis here (or take it out).

(12) One interesting result is that gradients, in addition to providing a global cue, also help establish polarity. Is there any experimental evidence to support that?

(13) In the analysis of mutant clones (Figure 6), the authors needed to add another term to the equations. It is really unclear why such a term has to be included. Why does it have to be included here but not in the section? Why does this term have to be in this form (3rd power of the ft polarity)? The authors need to justify these choices so that one can assess if this is legitimate or not.

(14) Figure 5a – explain color scheme.

(15) Figure 6 – why is the range of rho in the x-axis was chosen to be so small?

*Reviewer #2 (Recommendations for the authors):*

Here are detailed comments/questions regarding the article:

– p3 "Local interactions such as the formation of heterodimers have been considered as an output of the formation of polarity and used as a readout of polarization". In ref. 24 and 25, local interactions such as the formation of hetero-dimers seem required for cell polarization and do not seem its consequence.

– In figure 5 the effect of the gradient\epsilon seems rather weak while its effect on robustness (Figure S4) seems stronger. One of the main conclusions of the article seems also to be the effect of the gradient on robustness. Therefore, it seems necessary to move figure S4 from the SI to the article.

– In Figure 3, x goes from 0 to 500, while in Figure 5 ρ=0.5 and \epsilon=0.1. Does this mean that densities can be negative?

– In the introduction, the discussion about global and local modules is confusing: One may expect the global module to only concern tissue scale interactions while the local module would only relate to local interactions. A discussion explaining why this distinction between modules and how to interpret them would be useful.

– No reference to Figure 1 is made in the article. The figure is poorly explained and poorly understandable.

– In Eq 2-5 the use of a dot for time derivatives should be reminded.

– p7, Shouldn't it be \theta_i = 0 = 2 pi = pi/3, 2pi/3, pi, 4 pi/3, 5pi/3?

– p8, "The presence of threshold is due to the inter-cellular interactions and heterodimer formation between the proteins across the membranes of two neighboring cells(Figure 2 E)". The relation between the sentence and the referred panel seems unclear.

– Eq (25), the notation " (.)' " for spatial derivatives should be introduced.

– No reference is made to Figure 2A and B in the article.

– In Figure 2A, \β' does not correspond to the notations of the article.

– Figure 2A and elsewhere the use of red and blue stripes in cells is not explained.

– Figure 2B I don't understand the panel and why this is useful.

– Fig2F the x-axis seems in a log scale but this has not been clearly stated.

– The term "polarity" or "polarization" refers sometimes to a quantity defined by a cell scale and some other times to a quantity defined at the whole system scale making a clear distinction between the two would help the reader. Using different notations may also help.

– p11, the paragraph starting with "When the noise amplitude…" gives the impression that the noise only impacts the direction of cell polarization while FIGURE 3 C-E gives the impression that noise also changes the amplitude of cell polarization with regions with polarization p=0.

– p11, "First we observe aster and swirling patterns where polarization (P) in multiple cells either originate or terminate, or rotate around a point, respectively". If the result of simulations shows a random field of polarization, it does not seem so easy to identify patterns such as asters or swirling. To be convincing, the authors should indicate clearly the patterns they are able to identify.

–Eq (28), what is the physical justification (meaning) of the added term?

–Figure 6A, the color code has not been explained.

– Figure 7B What we see is not really explained and we ignore how to compare it whit numerical results. In the caption, the sentence "Cells around the deletion regions (blue) are polarized in the direction" misses an end.

– Figure 8ABCD, the orientation of the gradient is not clearly given. Panel B is not really explained and is difficult to relate to numerical results.

*Reviewer #3 (Recommendations for the authors):*

Some assumptions underlying the model need to be justified in more detail:

(1) Why is additive noise sufficient? How do you determine the amplitude of the additive noise? Is it really a free parameter?

(2) Eq.(1) What do you mean by writing that c is small? Is this the case for cells?

(3) Why did you choose to average your results over 50 realisations? Why not more or less?

(4) Section "Presence of gradient and local down-regulation explains the domineering non-autonomy": I do not understand this part. What is the effect that I should look for? I am not sure to get what you mean by "domineering non-autonomy" – I have some idea, but am not sure. Please explain. Also, I am not sure what I should take from Figure 8C, D. Just that the arrows are inverted compared to Figure 7C, D? The evidence you give seems to be anecdotal as you show essentially one example.

[Editors' note: further revisions were suggested prior to acceptance, as described below.]

Thank you for resubmitting your work entitled "Emergence of planar cell polarity from the interplay of local interactions and global gradients" for further consideration by *eLife*. Your revised article has been evaluated by Aleksandra Walczak (Senior Editor) and a Reviewing Editor.

The manuscript has been improved but there are some remaining issues that need to be addressed, as outlined below:

(1) Improve the structure of the manuscript such that going back and forth to the material and methods is not necessary to understand the work. In particular, references should not be made to points that are developed only later.

(2) Improve referencing to experimental results as detailed in your report. Particular attention should be paid to clearly distinguish between global (Ft/Ds) and core PCP pathways.

Please, consider also the detailed comments of Reviewer #1 below.

*Reviewer #1 (Recommendations for the authors):*

The manuscript by Singh et al. analyzes a new model for Ft-Ds mediated PCP, based on stabilization of membrane complexes. The original manuscript discussed the main idea of the model, but lacked novel predictions and some quantitative analysis. The revised version of the manuscript significantly expands on the first submission, and provides additional analysis of the model as well as some experimental predictions. There are however two main issues with the manuscript that are problematic: First, the structure of the current version makes it difficult to read and follow the logic and results in a coherent manner. Second, references to experimental data is problematic, as many of the results are compared to experiments in the Core pathway without clearly mentioning it (though it relies on a somewhat different molecular mechanism), and in some cases references to experimental results are misinterpreted or missing. There are other issues as well as detailed below. Somehow this submission seems sloppier than the previous one…Despite all these drawbacks, I believe that after proper editing and clarifying everything the manuscript can be useful to the community as an alternative model for PCP that can be tested experimentally.

The main issues:

(1) The structure of the manuscript makes it really hard to follow. The main issue is that one cannot understand the results without going back and forth to the material and methods. In the previous version the Materials and methods were a section right after the introduction, and the assumptions, definitions, and equations were presented there. In the current version, the Materials and methods section is at the end. The results start referring to the figures without properly introducing the model first. Since the model is the essence of the paper, it must be presented properly there. The material and methods should contain the detailed derivations. This of course will create some overlap, but it would be better than the current format. For example, the first two equations in the main text (eq 1,2) are very specific equations of a continuum model, which is not used for deriving almost any figure, and is only used to derive some analytical expressions (that may be useful, but are not the main focus of the paper). This makes the whole of the first section cryptic to the reader and confusing at best.

(2) There are several places where there are problematic references to other sections. For example, in row 170 it says: "We have incorporated static or "quenched" noise in the equations as described earlier (Eq (26))." But eq 26 is derived later in the manuscript…A similar thing happens again in row 179. Even within the Materials and methods there is a confusion. For example for the derivation of eqs. 15-18, within the Materials and methods, the reader is referred to the material and methods…(presumably to the Continuum limit section).

(3) The continuum model and the insights we learn from it are a bit cryptic due to the unorganized form by which it is presented. Equation 3 is interesting, but I believe that the form obtained is only for Hill coefficient m=2, right? It is really confusing to have the same letter, m, both as an index of membrane and as an index for Hill coefficient.

(4) Another main issue is the problematic referencing to experimental results. First, there are two places where the references are absent: Figure 4C (both in the text and in the caption) and Figure 10A. Second, some references are misleading or problematic. In Figure 9 There is a comparison between the model (B and C) and the results (A). The model shows non-local polarization defect around a clone lacking Ft or overexpressing Ft. However, the data shows that Ds around the clone to be more diffusive rather than polarized (as mentioned in the original paper: "Loss of ft causes Ds to be no longer tightly localized at cell boundaries and appears to result in a more diffuse subcellular protein distribution within the cell. "), the model predicts changes in polarity.

(5) In fact, most of the experimental panels the model compares to show polarity patterns from the core pathway and not the Ft-Ds pathway. This is rather troubling since it is unclear if the model is suitable for the core pathway. At least, it should be clear what panels are from the Core pathway and what panels are from the Ft-Ds. The authors argue that complex stability may play a role in the core pathway, however, it is clear that this is not the main mechanism there since there are cytoplasmic proteins which convey the polarity (Pk, Disheveled). Hence, the authors should: (1) clearly state when referring to Core vs Ft/Ds experiment. (2) Add a section in the discussion about the limitation of the model when applied to the core pathway.

(6) Line 226: "This interesting finding accounts qualitatively for the swirling patterns observed in experimental systems by Axelrod et al. (2020). This reference is a general review…unclear how it is related to the results.

(7) The analysis of swirling is not very convincing (Figure 4 Figure supp 2). These figures show the curl and divergent of several simulations. Would be nice to get relevant length scales from these or some other quantitative measure.

(8) The schematic in Figure 1C is not very clear. It is not clear which proteins are on the membrane and which ones are on the membrane. The color scheme is not clear and not specified

(9) Line 140 – the reference to the figure is not in the right line.

(10) Figure 5 – Why are the images blurred? It seems like an effect that hides the fact that the correlation length jumps very abruptly from 0 to 1. Is that the case? Can the authors provide a clearer picture of what happens at the transition interface?

(11) Why is there a cusp in Figure 8b?

(12) Page 8: "and at points that are 5,10,50 cells away from the deletion boundary". In the figure the values are different

(13) Discussion page 9. " We are left with only 3 parameters…". There are also the Hill function parameters no?

(14) The new discussion is quite nice as it relates the model to specific system behavior and predictions.

---

## [Author Response]

Essential Revisions:(1) Several models with a similar hypothesis and similar predictions have been published previously. Clearly point out in which way your mechanism fundamentally differs from these previous works. Also discuss, in which way your mechanism is better than those studied in previous works. In particular, explain features that are not explained by other models, and/or generate non-trivial predictions that can be tested experimentally.

In PCP literature, the modeling works can be broadly classified into two types. First, the models with mechanistic details, which take into account the concentrations of specific proteins of core or global modules of PCP along the cell boundary, using approaches such as mass action kinetics. Second, the phenomenological class of models which are relatively less detail-intensive and focuses on understanding the role of protein-protein interactions on PCP establishment. In both of these modeling approaches there is some feedback mechanism which is usually described in terms of the Ft-Ds heterodimers (in the case of models studying global module of PCP) or Fmi-Fmi (+ other cytoplasmic proteins) homodimers (for the core module). As a result of these feedback mechanisms, the cell polarity is, therefore, defined in terms of the asymmetric distribution of the inter-cellular dimers of proteins/protein-complexes. The heterodimer-based feedback mechanism inherently assumes protein-protein interactions at the cytoplasmic ends (*cis* inhibition) along with that at the extracellular end of the proteins. Recent works with in vitro synthetic cell systems have demonstrated that in the co-culture of cells expressing only Ft and only Ds, these proteins get localized at the cell-cell interface without any intracellular Ft-Ds inhibition. This provides another justification for the only inter-cellular interactions in the global module of PCP.

In the model presented in this work, we do not consider intra-cellular interaction among the proteins of the global module of PCP, and therefore, demonstrate that PCP can be established via inter-cellular interactions among proteins at the extracellular ends only. This simplification results in a minimal model with only two independent parameters (in the absence of expression gradient).

We have comprehensively revised the sections on the modeling details and discussion to address this concern. We have also added a new section to present a summary of the previous modeling works on PCP (section 2.1). Additionally, in the Discussion section of the manuscript, we have also proposed new experiments which can help ascertain the basic assumptions of the proposed model and those of the earlier ones.

In the discussion, you suggest two predictions: One is to measure the decay length around a clone. This is not really a prediction. The other is to look at the asymmetry around a Fat4 clone in the presence of a gradient. This can be interesting, but it was not analyzed in the manuscript, nor was there a serious effort to check if this was observed experimentally.

In the revised version of the manuscript we are presenting quantification of the results, earlier shown in a qualitative manner, including the decay length and asymmetric non-autonomous polarity. We have also revised the text describing the effect of the local loss of Ft or Ds in the presence of Ft gradient and discussed some experimental works showing non-autonomous effects of Ft loss. We thank the reviewer for pointing out the lack of experimental observation of the effects of the localized Ft/Ds loss which we have now included in the revised manuscript. The experimental images are included in Figures 9 and 10. Further, we have also added a new section in the Discussion to describe the specific model predictions and potential experiments for their validation.

(2) Some conclusions of this work rely on visual appreciation rather than quantification. This is particularly true for what concerns 2D patterns. An argument of the authors is for example that their model reproduces a variety of known spatial patterns, but the comparison with experiments is only visual. Provide a more quantitative comparison.

In the revised version we have produced the results in a quantified manner, especially those from the 2D model. We have added new figures (Figures 5, S2) which present the quantification of model predictions. In addition, we have also added new panels in the existing Figure 10 to quantify the effect of domineering non-autonomy due to the localized loss of Ft and Ds. Furthermore, in this work, we also present a *continuum theory* (in one dimension, details are given in supporting information) along with the discrete *computational* model. The continuum theory does not only recapitulates the discrete computational simulation results but also provides a comprehensive understanding of the dependence of the system parameters (such as protein levels *ρ*, binding-unbinding rates *α* and *β*, gradient level *E*, distance dependence in domineering non-autonomy, etc.) on PCP levels in the tissue. Even though there are several *computational* models for PCP in epithelial tissues, we believe, this is the first attempt to have a detailed continuum theory for PCP.

(3) Justify better some assumptions underlying the model. Notably, why should Fat-DS complexes be stabilized when bound, and why is additive noise appropriate to account for the effect of stochasticity in the attachment-detachment dynamics?

We have modified the sections on the assumptions of the model and given justification for each in respective sections. The changes concerning the questions raised here can be seen in sections – 2.1, 2.2, 2.3, and 2.5 (not exhaustive). In particular, the assumption of the stabilization of the Ft-Ds complex, which is reflected in a decrease in the detachment rates of Ft and Ds, can be justified from the in vitro experimental observations in the work by Loza et al. In this work, the Ft and Ds get asymmetrically localized at the interface of the synthetic cells having Ft and Ds. In the absence of the interaction between the cells (isolated cells) Ft and Ds are localized isotropically. Regarding the additive nature of the noise in the attachment-detachment dynamics: The noise variance in kinetic equations for chemical concentrations scales as the inverse of the mean number of molecules involved and is thus in general multiplicative in nature. If we replace these numbers with their mean values the noise is rendered additive. Such an approximation should be qualitatively correct for small fluctuations in a stationary state with well-defined mean values. This is similar to the approach of S Shadkhoo & M Mani, PLoS Comput Biol 15, e1007454 (2019)

Reviewer #1 (Recommendations for the authors):More specifically I have the following major comments:(1) The main claim of the authors is that the new model is 'better' than previous models since it is a relatively simple phenomenological model, yet it focuses on the relevant boundary proteins (and not complexes). While I appreciate the elegance of the new model, it is unclear in what sense is it better. I think the important measure for this question should be its ability to explain experimental behaviors that were not explained before, or generate non-trivial predictions that can be put to experimental test.

The mathematical model of the PCP presented in this paper follows a semi-phenomenological approach with “protein-clustering” based mechanism of molecular interactions. Cell membranes are known to be laterally organized into domains of specific protein compositions (usually via intracellular interactions among proteins) [Radhakrishnan et al., Ann. Biomed. Engg., 2013; Lin et al., Biophys. J. 2018; Sakamoto et al., 2023 PNA Nexus]. In this work, we have extended this mechanism to inter-cellular protein-protein interactions which can result in asymmetric clustering of proteins on cell membrane. We have shown that the mutual affinity of the two proteins at the cell junctions can result in their asymmetric localization without explicit consideration of the heterodimer formation.

This approach is in contrast to the existing mathematical descriptions of PCP where heterodimer formation at the junctions of two neighboring cells is often an indispensable model ingredient.

In the revised version of the manuscript, we have highlighted these differences in the basic model assumptions (see Section 2.1). Further, we have also proposed experiments to differentiate the basic assumptions of the proposed models and others using experimental methodologies of few recent works (such as Medina et al., 2023 Nature Comm. and Loza et al., 2017 *eLife*) in Discussion.

Despite these simpler assumptions, the model is not only capable of predicting most of the known features of the planar cell polarity but also makes quantifiable predictions that can be tested. We list some of these below

Effect of attachment rate of proteins to the membranes on PCPEffect of binding affinities of Ft and Ds to each otherQuantification of correlation length of PCP which has neither been estimated by any previous model nor measured in an experiment.Effect of tissue-level protein expression gradients in enhancing the stability of PCPPCP decay length from any edge such as veins in *Drosophila* wing or tissue boundaries.

These predictions have been described in detail in the Discussion section of the manuscript along with possible ways for their experimental realizations.

In the discussion, the authors suggest two predictions: One is to measure the decay length around a clone. This is not really a prediction.

Domineering non-autonomy, in which signalling mutants disrupt polarity nonlocally in neighbouring wild-type cells, has been a well known phenomenon in PCP. However, as far as we know, the effect has been reported in all of the experimental as well as modeling literature using *qualitative* descriptions. In this work, we have performed quantitative analysis of decay length around a clone (Figure 10D-E) to identify its dependence on the protein levels *ρ* and their binding affinities. The exponential decay predicted by the model can be tested using experiments which can provide a quantitative measure of domineering non-autonomy.

The other is to look at the asymmetry around a Fat4 clone in the presence of a gradient. This can be interesting, but this was not analyzed in the manuscript, nor was there a serious effort to check if this was observed experimentally. In summary, the model should not only be elegant but should have non-trivial consequences.

We have looked at the localized loss of Ft and Ds in the presence as well as absence of tissue level expression gradients (Figures 9 and 10). In both cases the nonlocal effects of protein loss/overexpression were seen. It has been correctly pointed out by the reviewer that the effects of localized loss of Ft has not been conclusively tested experimentally due to the role played by Ft in tissue growth. It has, however, been seen in case of localized loss of Ds (for example in Strutt and Strutt, 2002 Dev. Cell) where nonlocal effects in Ft localization were seen around *ds* clones. We have included this referece in the corresponding figure.

(2) I find it interesting that the assumption that Fat-Ds complexes are stabilized is sufficient to promote global polarity. Most models to date, rely on feedback where complexes in one direction promote the formation of complexes in the same direction and prevent complexes in the opposite direction (Mani et al). The stabilization hypothesis effectively promotes complexes in the same direction (since more Fat and Ds remain bound). However, it is unclear why such interactions can promote complexes in both directions in parallel. I suspect it has something to do with the finite pools of Fat and Ds, but it would be nice to get an intuitive explanation for that.

We thank the reviewer for pointing this out. The finite availability of the two proteins in the cell indeed has a role in establishing the asymmetry in the protein levels in the cells. In order to test this explicitly, we have performed some additional simulations where the rate of protein binding to the membrane does not depend on the protein availability. This is equivalent to having an infinite pool of both proteins in the cells.

This results in a modified form of equations (2)-(5) which is

*f*˙*l*(*i*) = *αf_∞_*(*i*) *− β*_0_*f_l_*(*i*)*H*_2_(*d_r_*(*i −* 1)) + *L_f_* (*i, t*)

*f*˙*r* (*i*) = *αf_∞_*(*i*) *− β*_0_*f_r_*(*i*)*H*_2_(*d_l_*(*i* + 1)) + *R_f_* (*i, t*)

*d*˙*l*(*i*) = *αd_∞_*(*i*) *− β*_0_*d_l_*(*i*)*H*_2_(*f_r_*(*i −* 1)) + *L_d_*(*i, t*)

*f*˙*r* (*i*) = *αd_∞_*(*i*) *− β*_0_*d_r_*(*i*)*H*_2_(*f_l_*(*i* + 1)) + *R_d_*(*i, t*)

where *f_∞_*(*i*) and *d_∞_*(*i*) take some constant values. In this condition, we observe a steady state with uniform protein levels on the cell membrane, *i.e.* failure of the cell polarity establishment. We have discussed this aspect in section 3.1 and also added Figure S1-C to illustrate this.

(3) Just to point out that there are papers that do show enhanced stability of Fat-Ds complexes, and it would be nice to mention these: Hale et al., eLife 2015, Loza et al., eLife 2017.

We thank the reviewers for pointing this out. We have mentioned these works in the Introduction and Methods section of the revised manuscript.

(4) When providing images from published work – please provide mutant details (for example in the PCP mutants in Figure 1).

We have added the mutant details in figures with experimental images.

(5) Figure 2 – please provide a plot showing one realization of the simulation showing the local polarity in each cell. It would help clarify the behavior of the local polarity. In 2F, the axis uses peculiar tick numbers. Unclear if the axis is linear or logarithmic.

We have added a new panel in Figure 2 to illustrate the local polarity in a few cells. Additionally, the concentration of Ft and Ds on the right and left side of the cell as a function of time is also shown in Figure S1. The scaling in Figure 2D (which was earlier Figure 2F) has been modified along with the addition of the curve showing analytically calculated boundary between ‘polarized’ and ‘unpolarized’ regions in the phase diagram.

(6) Section on noise – Please compare to Burak et al. Are the results different/similar?

In this work, we have looked at two sources of noise: first, a Gaussian white dynamic noise in the protein kinetics (equations 9-12), and second, a static or quenched noise due to time-independent variability in total protein levels (equations 15) in the cells. Burak & Shraiman, PLoS Comp. Bio. 5, e1000628 (2009) focus on the effect of stochasticity in protein kinetics, in the form of a Gaussian temporally white noise with concentration-dependent strength due to Poisson statistics. Both studies demonstrate the emergence of swirling patterns in the absence of tissue-level expression gradient.

In both of these noise scenarios, we have also quantified the correlation length of the tissue polarization and its dependence on the protein levels and noise strengths, which has been lacking in all previous works, as far as we know. We have elaborated the comparison of the effect of stochasticity on PCP between this work and that of Burak & Shraiman in section 3.2.

(7) Figure 3B-E – the x-axis describes tissue position. Please clarify what is the scale of one cell.

The tissue position or lengths in the discrete and continuum models are represented in terms of characteristic cell size. This has been added in the caption of Figure 3, section – 2.7, and also in Appendix A. We have also changed the axis labels in Figure 3 to indicate cell index instead of tissue position.

(8) Figure 4A – The color scheme is unclear. Specify what is blue and what is red, as well as what the dimensions of each domain (red, white, blue) within the cells refer to.

We have changed the color scheme of the schematic to avoid confusion at all places in the manuscript. We have also labeled the colors and added explanations for these schematics in respective figure captions.

(9) In Figure 4C it seems that some domains are very large and others are rather small. This is rather different than the swirling observed in some PCP mutants. Can the authors discuss this discrepancy (does it depend on parameters?)?

Firstly, Figure 4C has been revised with the modifications in the model which demonstrates qualitatively similar swirling patterns as those for the mutants. Secondly, the nature of the swirling patterns indeed depends on the strength of the stochasticity. For very low strength of the noise, the planar cell polarity is more or less coordinated throughout the tissue. For high noise strengths, however, the domains of coordinated tissue polarity reduce in size. This not very surprising effect also results in a decrease in the overall tissue-polarity magnitude with an increase in the noise strength (Figure 3).

To study this aspect quantitatively we also calculated the polarization correlation length as a function of protein levels and noise strengths. These new calculations are shown in the newly added Figure 5 in the revised manuscript.

(10) Figure 4D. The author refers to a manuscript by Michael Simon, but this picture does not appear to be from that paper. As far as I can tell from the image, this is an overexpression of Fat that leads to swirling. Can the authors check if their model captures this behavior?

It is indeed correct that image in Figure 4D is for Ft overexpression and is from a different publication. We apologize for this oversight and this error has been fixed. The overexpression of Ft (or any other membrane-bound PCP protein, for that matter) results in the loss of the tissue level gradient and can also introduce inhomogeneity (akin to static noise) in the protein expressions at the tissue scale. As shown in this figure, such protein overexpression, therefore, results in swirling patterns in the tissue as long as the average protein levels in the cells are above the critical threshold for polarization.

(11) The authors refer several times to another submitted manuscript (ref 38). One cannot provide results based on unpublished work. Either provide a reference or provide the full analysis here (or take it out).

The reference to this work has been removed, and all relevant analytical details have been included as supporting information in the revised manuscript.

(12) One interesting result is that gradients, in addition to providing a global cue, also help establish polarity. Is there any experimental evidence to support that?

Unfortunately no experimental manipulation of expression gradient has been realized, as far as we know. Note that the result, albeit interesting, is not very surprising given the similarities of the PCP model with that of well-studied models in classical statistical mechanics, such as those for magnetization. In magnetization, the external magnetic field (analogous to the expression gradients of PCP proteins) not only provides the global cue but also helps in magnetization. This also presents one possible experimental test where protein levels are reduced to below the critical levels for polarization but are present in a tissue scale gradient. We have discussed this experimental setup in the Discussion section of the manuscript.

(13) In the analysis of mutant clones (Figure 6), the authors needed to add another term to the equations. It is really unclear why such a term has to be included. Why does it have to be included here but not in the section? Why does this term have to be in this form (3rd power of the ft polarity)? The authors need to justify these choices so that one can assess if this is legitimate or not.

The cubic term was added in the equations for the case of mutant clones to avoid any blowups when 1 *− γc <* 0 where *c* is the protein concentration on the membrane. This specific condition also implies that the protein unbinding rate turns negative at high protein levels. To address this systematically, we have modified expression for the protein unbinding from the membrane by using Hill’s function 11+(yc)m which was earlier modeled as a linear feedback 1 *− γc*. With this modification 11+(yc)m the unbinding rate always remains positive. Therefore, any additional term is no longer required for the study of the mutant clones. We would like to point out here that with this change in the detachment dynamics, we have revised all the results shown in the figures. It has to be noted that this change does not affect the qualitative nature of the results presented in the earlier version of the manuscript.

(14) Figure 5a – explain color scheme.

The colors in all schematics represent Ft and Ds and this has been mentioned in all figures.

(15) Figure 6 – why is the range of rho in the x-axis was chosen to be so small?

The figure has been revised to include a larger range of *ρ*.

Reviewer #2 (Recommendations for the authors):Here are detailed comments/questions regarding the article:

– p3 "Local interactions such as the formation of heterodimers have been considered as an output of the formation of polarity and used as a readout of polarization". In ref. 24 and 25, local interactions such as the formation of hetero-dimers seem required for cell polarization and do not seem its consequence.

It has been correctly pointed out that in the cited references, the local interactions via heterodimer formation have been considered essential for polarity establishment. This paragraph has been rewritten to correct this error.

– In figure 5 the effect of the gradient\epsilon seems rather weak while its effect on robustness (Figure S4) seems stronger. One of the main conclusions of the article seems also to be the effect of the gradient on robustness. Therefore, it seems necessary to move figure S4 from the SI to the article.

The effect of the gradient on the robustness of the PCP is indeed strong as compared to that on the magnitude of the polarization. We have analyzed this effect more thoroughly in the revised manuscript. In addition to the effect of the gradient in the presence of dynamic noise (shown previously in Figure S4), we also looked at its effect in the presence of static noise. The new results are presented in the revision in Figure S7. Figure S4 has also been moved to the main part of the manuscript as Figure 7.

– In Figure 3, x goes from 0 to 500, while in Figure 5 ρ=0.5 and \epsilon=0.1. Does this mean that densities can be negative?

We thank the reviewer for highlighting this. In all simulations with protein expression gradient, the reference tissue location (*x* = 0 in the continuum, and cell index *i* = 0 in discrete) was set at the two extremes (left or right for *E >* 0 and *E <* 0, respectively) of the simulation domain. This is to ensure that protein levels do not become negative in any of the cells. We have added this detail in section 3.4.

– In the introduction, the discussion about global and local modules is confusing: One may expect the global module to only concern tissue scale interactions while the local module would only relate to local interactions. A discussion explaining why this distinction between modules and how to interpret them would be useful.

We have rewritten this part of the introduction to remove any ambiguities. One point needs some emphasis here. Due probably to historical reasons, the Ft-Ds system is known as ‘global’ in nature in the context of PCP but its mechanism of action is also ‘local’ in the sense that Ft and Ds asymmetrically localize on the cell membrane due to local inter-cellular interactions. We have also highlighted this in the Introduction section.

– No reference to Figure 1 is made in the article. The figure is poorly explained and poorly understandable.

We regret this oversight. We have added references to each panel of Figure 1 at appropriate locations in the Introduction, Modeling, and Results section.

– In Eq 2-5 the use of a dot for time derivatives should be reminded.

We have added a sentence to this effect. Similarly, in the supporting information similar definition has also been added for the spatial derivatives.

– p7, Shouldn't it be \theta_i = 0 = 2 pi = pi/3, 2pi/3, pi, 4 pi/3, 5pi/3?

It is indeed correct that the summation in equations (20)-(23) is over discrete values of *θ_i_*. The equations have been modified and the specific values of *θ_i_* have been mentioned after equation (21).

– p8, "The presence of threshold is due to the inter-cellular interactions and heterodimer formation between the proteins across the membranes of two neighboring cells(Figure 2 E)". The relation between the sentence and the referred panel seems unclear.

It was a typographical error which has been fixed now.

– Eq (25), the notation " (.)' " for spatial derivatives should be introduced.

This equation is modified in the main text and the spatial derivative is not there. The equation with spatial derivatives is mentioned in the supporting information where the notation has been explained.

– No reference is made to Figure 2A and B in the article.

These figure panels have been removed in the revisions.

– In Figure 2A, \β' does not correspond to the notations of the article.

The schematic from (earlier) Figure 2A has been moved to Figure 1C and the notations have been corrected.

– Figure 2A and elsewhere the use of red and blue stripes in cells is not explained.

The color scheme in all figure schematics has been changed. It has also been explained in respective figure captions.

– Figure 2B I don't understand the panel and why this is useful.

These figure panels have been removed now.

– Fig2F the x-axis seems in a log scale but this has not been clearly stated.

The x-axis in the figure has been fixed now.

– The term "polarity" or "polarization" refers sometimes to a quantity defined by a cell scale and some other times to a quantity defined at the whole system scale making a clear distinction between the two would help the reader. Using different notations may also help.

We thank the reviewer for noticing the discrepancies. We have corrected these in the manuscript now. Whenever we use *p*(*i*) or *p*(*x*) with a positional argument it means the polarity of the cell with index *i* or at location *x*, whereas *p* (without any positional argument) refers to the average polarization in the tissue, as defined in equations 8 and 24.

– p11, the paragraph starting with "When the noise amplitude…" gives the impression that the noise only impacts the direction of cell polarization while FIGURE 3 C-E gives the impression that noise also changes the amplitude of cell polarization with regions with polarization p=0.

It is indeed correct that the noise results in not only change in the polarity direction but also the magnitude of polarization in some cells. In fact, introduction of noise results in formation of patches of cells of opposing polarity and with an increase in the noise amplitude the number (size) of these patches increases (decreases). The cells at the interface between any two patches of opposite polarity, therefore, gets affected by the polarization of its neighbors resulting in a modulation of its own polarity magnitude. We thank the reviewer for pointing this out and we have revised the description of this result in section 3.2.

– p11, "First we observe aster and swirling patterns where polarization (P) in multiple cells either originate or terminate, or rotate around a point, respectively". If the result of simulations shows a random field of polarization, it does not seem so easy to identify patterns such as asters or swirling. To be convincing, the authors should indicate clearly the patterns they are able to identify.

We have tried to quantify these patterns and calculated the correlation length as a function of total protein concentration and noise amplitude (Figure 5). We have also added a new figure with different realizations of PCP (Figure S5). In this figure, asters and swirls are highlighted (qualitative). Quantitatively, asters and swirls correspond to the regions of high values of divergence (∇*.*) and curl (∇*×*) of the PCP field. For the PCP vector field, we calculated its divergence and curl for the realization shown in Figure S5, and the plots are presented in Figure S6.

– Eq (28), what is the physical justification (meaning) of the added term?

This extra term was added earlier to remove any blowups when 1 *− γc <* 0 where *c* is the protein concentration on the membrane. As mentioned in the reply to comment#13 of Reviewer#1, due to the modifications in the model to better capture the effect of Ft and Ds unbinding from the cell membrane using Hill’s function 1 *− γc* is replaced with 11+(yc)m which is always positive. Therefore, the extra term is no longer required for the study of the mutant clones.

– Figure 6A, the color code has not been explained.

The colors in the schematic have been changed to avoid confusion with other colors in the figures. In the revised version, descriptions of the color codes have also been added in respective figures.

– Figure 7B What we see is not really explained and we ignore how to compare it whit numerical results. In the caption, the sentence "Cells around the deletion regions (blue) are polarized in the direction" misses an end.

Figure 9 (earlier Figure 7) has been modified and the nature of the perturbations in terms of the protein downregulation/upregulation has also been highlighted. The section pertaining to this figure in the main text has also been modified to describe each panel of the figure. Similarly, we have also rewritten the section 3.6 to explain the results presented in the (new) Figure 10.

– Figure 8ABCD, the orientation of the gradient is not clearly given. Panel B is not really explained and is difficult to relate to numerical results.

The tissue scale variations in protein levels have been denoted schematically in Figures 6, 9, and 10.

Reviewer #3 (Recommendations for the authors):Some assumptions underlying the model need to be justified in more detail:(1) Why is additive noise sufficient? How do you determine the amplitude of the additive noise? Is it really a free parameter?

The stochasticity in the protein binding-unbinding kinetics is due to the molecular noise in these biochemical reactions. The noise variance in the corresponding Langevin equation should depend on the molecular concentrations and is therefore multiplicative. If we make the approximation of replacing these numbers by their mean values the noise is rendered additive. Such an approximation should be qualitatively correct for small fluctuations in a stationary state with well-defined mean values. This is similar to the approach of S Shadkhoo & M Mani, PLoS Comput Biol 15, e1007454 (2019). We have not attempted to estimate noise magnitudes, but have introduced noise terms with small strengths to observe trends. The variance of the noise scales as the inverse of the total number of molecules on the interface at which the binding kinetics is taking place [Burak & Shraiman PLoS Comput Biol 2009, or S Shadkhoo & M Mani, PLoS Comput
Biol 2019]. Thus, though it is not a free parameter, it is hard to tie down precisely.

(2) Eq.(1) What do you mean by writing that c is small? Is this the case for cells?

We have modified the equations by substituting Hill’s function for 1 *− γc* (section 2.3). For the small value of *c*, however, we obtain the same equation as presented in the earlier version of the manuscript. With this change, the resulting equations do not require *c* to be small.

(3) Why did you choose to average your results over 50 realisations? Why not more or less?

In the presence of noise, we found that for 50 realizations the average and standard deviation values of cell/tissue polarity, membrane-bound protein levels, etc. demonstrated convergence. Therefore, the results shown in Figures 3A, 5, 7, and S3A the values are shown as average and standard deviations over 50 realizations.

(4) Section "Presence of gradient and local down-regulation explains the domineering non-autonomy": I do not understand this part. What is the effect that I should look for? I am not sure to get what you mean by "domineering non-autonomy" – I have some idea, but am not sure. Please explain. Also, I am not sure what I should take from Figure 8C, D. Just that the arrows are inverted compared to Figure 7C, D? The evidence you give seems to be anecdotal as you show essentially one example.

The phenomenon of domineering non-autonomy in PCP is the disruption of polarity in the wild-type cells that are close to a boundary, such as that of a *ft^−/−^* clone. It has been extensively reported in the context of the disruption in the protein levels of the core module of PCP (see reviews by Devenport, 2014, J. Cell Bio, or Strutt and Strutt, 2021, Royal. Soc. Open Biology for details). We have rewritten the description of the results on this aspect in section 3.6. We have also added new panels in Figure 10 (panels D and E) to quantitatively demonstrate this effect. Unfortunately, almost all works (including experimental and modeling) have studied domineering non-autonomy with visual and qualitative evidence. We hope that similar quantification (as shown in Figure 10D-E) can also be performed on experimental images to look at the competitive effects of expression gradients and local protein loss.

[Editors' note: further revisions were suggested prior to acceptance, as described below.]

The manuscript has been improved but there are some remaining issues that need to be addressed, as outlined below:(1) Improve the structure of the manuscript such that going back and forth to the material and methods is not necessary to understand the work. In particular, references should not be made to points that are developed only later.

We have restructured the manuscript to address this. In the revision, the mathematical model of PCP is described before the results. The detailed calculations of the specific cases of the continuum model have been shifted to appendices.

(2) Improve referencing to experimental results as detailed in your report. Particular attention should be paid to clearly distinguish between global (Ft/Ds) and core PCP pathways.

In the Results section of the revised manuscript, all the experimental images are from the works perturbing the members of the global (Ft/Ds) module of PCP. The original references for these images have been cited in the figure caption along with the main text. In the Discussion section, we have also discussed the possible extension of the proposed model to the core pathway. The limitations of the current version of the model for the core pathway are also discussed.

Please, consider also the detailed comments of Reviewer #1 below.Reviewer #1 (Recommendations for the authors):The manuscript by Singh et al. analyzes a new model for Ft-Ds mediated PCP, based on stabilization of membrane complexes. The original manuscript discussed the main idea of the model, but lacked novel predictions and some quantitative analysis. The revised version of the manuscript significantly expands on the first submission, and provides additional analysis of the model as well as some experimental predictions. There are however two main issues with the manuscript that are problematic: First, the structure of the current version makes it difficult to read and follow the logic and results in a coherent manner. Second, references to experimental data is problematic, as many of the results are compared to experiments in the Core pathway without clearly mentioning it (though it relies on a somewhat different molecular mechanism), and in some cases references to experimental results are misinterpreted or missing. There are other issues as well as detailed below. Somehow this submission seems sloppier than the previous one…Despite all these drawbacks, I believe that after proper editing and clarifying everything the manuscript can be useful to the community as an alternative model for PCP that can be tested experimentally.

We thank the reviewer for valuable comments. We have restructured the manuscript to address the concern about its organization and changed some of the experimental results presented in the figures. Specific changes are listed in the responses below.

The main issues:(1) The structure of the manuscript makes it really hard to follow. The main issue is that one cannot understand the results without going back and forth to the material and methods. In the previous version the Materials and methods were a section right after the introduction, and the assumptions, definitions, and equations were presented there. In the current version, the Materials and methods section is at the end. The results start referring to the figures without properly introducing the model first. Since the model is the essence of the paper, it must be presented properly there. The material and methods should contain the detailed derivations. This of course will create some overlap, but it would be better than the current format. For example, the first two equations in the main text (eq 1,2) are very specific equations of a continuum model, which is not used for deriving almost any figure, and is only used to derive some analytical expressions (that may be useful, but are not the main focus of the paper). This makes the whole of the first section cryptic to the reader and confusing at best.

We agree with the reviewer that having modeling details at the end of the manuscript makes it difficult to comprehend the work. We have restructured the article by moving the model details before the results. The detailed derivations of the specific results are kept in the appendices at the end of the article. With this change, we have also addressed the specific instances mentioned in the comment. We hope this will make it easier to follow the mathematical model proposed in this manuscript.

(2) There are several places where there are problematic references to other sections. For example, in row 170 it says: "We have incorporated static or "quenched" noise in the equations as described earlier (Eq (26))." But eq 26 is derived later in the manuscript…A similar thing happens again in row 179. Even within the Materials and methods there is a confusion. For example for the derivation of eqs. 15-18, within the Materials and methods, the reader is referred to the material and methods…(presumably to the Continuum limit section).

As mentioned in the response to the previous comment, we have rearranged the manuscript’s text. We have ensured that such problematic references are no longer present.

(3) The continuum model and the insights we learn from it are a bit cryptic due to the unorganized form by which it is presented. Equation 3 is interesting, but I believe that the form obtained is only for Hill coefficient m=2, right? It is really confusing to have the same letter, m, both as an index of membrane and as an index for Hill coefficient.

It is correct that the expression mentioned in the comment is derived for *m* = 2 (notation changed to *k* = 2 in the revision). One can derive the same for other values of *k* as well. Since the qualitative nature of the results does not depend on values of *k*, we have presented all the results (from continuum and numerical simulations) for *k* = 2. We have discussed this in the section “Parameter values”.

(4) Another main issue is the problematic referencing to experimental results. First, there are two places where the references are absent: Figure 4C (both in the text and in the caption) and Figure 10A. Second, some references are misleading or problematic. In Figure 9 There is a comparison between the model (B and C) and the results (A). The model shows non-local polarization defect around a clone lacking Ft or overexpressing Ft. However, the data shows that Ds around the clone to be more diffusive rather than polarized (as mentioned in the original paper: "Loss of ft causes Ds to be no longer tightly localized at cell boundaries and appears to result in a more diffuse subcellular protein distribution within the cell. "), the model predicts changes in polarity.

In the result section of the manuscript, we have presented experimental images (taken from other sources) in three figure panels.

(a) Figure 4(C): We are not sure why the reviewer has found references to be missing. The experimental research paper (Ambegaonkar and Irvine, 2015, *eLife*) from which the experimental image has been taken has been cited in the figure caption as well as in the main text (line 379 in the revision).

(b) Figure 9(A): We have replaced this panel with an experimental image from Sharma and Mcneill, 2013, Development which shows a reversal in the polarity direction in a non-local manner on clonal perturbation in Ft expression.

(c) Figure 10(A): In this panel, the experimental research paper (Chorro et al., 2022, Open Biology) was cited in the figure caption but was missing from the main text. We have added the citation in the main text as well (line 479 in the revision).

(5) In fact, most of the experimental panels the model compares to show polarity patterns from the core pathway and not the Ft-Ds pathway. This is rather troubling since it is unclear if the model is suitable for the core pathway. At least, it should be clear what panels are from the Core pathway and what panels are from the Ft-Ds. The authors argue that complex stability may play a role in the core pathway, however, it is clear that this is not the main mechanism there since there are cytoplasmic proteins which convey the polarity (Pk, Disheveled). Hence, the authors should: (1) clearly state when referring to Core vs Ft/Ds experiment. (2) Add a section in the discussion about the limitation of the model when applied to the core pathway.

As mentioned in the response to the previous comment, the comparison of the model with the experimental works are shown with perturbations in the Ft/Ds module only (Figure panels 4(C), 9(A), and 10(A)). In the Discussion section, we have added the limitations of the current version of the model if applied to the core module.

(6) Line 226: "This interesting finding accounts qualitatively for the swirling patterns observed in experimental systems by Axelrod et al. (2020). This reference is a general review…unclear how it is related to the results.

We have revised this part to include specific references.

(7) The analysis of swirling is not very convincing (Figure 4 Figure supp 2). These figures show the curl and divergent of several simulations. Would be nice to get relevant length scales from these or some other quantitative measure.

In the absence of the gradient cue we observe non-homogeneous polarity patterns. These swirling polarity patterns are composed of vortices (quantified by ∇*×*) and asters (quantified by ∇*·*) but their positioning and strength are strongly dependent on the initial conditions. Also, it is not possible to experimentally generate patterns of only vortices or only asters. Therefore, a quantitative comparison of the curl and divergence of the polarity field with those in experiments is very difficult. Therefore, we calculated the length scales of the polarity correlation as a function of total protein levels *ρ* and noise strengths *S* and *η*. These plots are shown in Figure 5. These quantities can be compared with experimental observation with the help of quantitative image analysis. We have mentioned this experimental validation of the model in the Discussion section.

(8) The schematic in Figure 1C is not very clear. It is not clear which proteins are on the membrane and which ones are on the membrane. The color scheme is not clear and not specified

We have revised Figure 1C schematic to address this concern.

(9) Line 140 – the reference to the figure is not in the right line.

Unfortunately, we are unable to understand this comment by the reviewer. In line 140 (previous version), we have presented the continuous nature of the transition of the tissue from an unpolarized to a polarized state. We have shown this by plotting the order parameter *p* against ∆*ρ* = *ρ − ρ_c_* which shows *p ~* ∆*ρ*^1*/*2^, a signature of a continuous phase transition (shown in Figure 2; figure supplement 2).

(10) Figure 5 – Why are the images blurred? It seems like an effect that hides the fact that the correlation length jumps very abruptly from 0 to 1. Is that the case? Can the authors provide a clearer picture of what happens at the transition interface?

The plot in Figure 5 is constructed by numerical simulation of the model for different combinations of *ρ* and noise magnitudes *S* and *η*. The transition shown in the figure is a continuous one. Since the correlation length is normalized by the system size and we have used a system of size 20 *×* 20 for the simulation, it gives the impression of an abrupt increase from 0 to 1. Values between those two extremes can, however, clearly be seen in the figure.

(11) Why is there a cusp in Figure 8b?

The polarization decay length in this figure diverges at *ρ* = *ρ_c_*. Its appearance as a cusp is due to the finite resolution in the *ρ* values used in the figure. We have revised the figure to remove the appearance of the cusp.

(12) Page 8: "and at points that are 5,10,50 cells away from the deletion boundary". In the figure the values are different

We have revised the text to match the plots shown in the figure.

(13) Discussion page 9. " We are left with only 3 parameters…". There are also the Hill function parameters no?

This statement has been revised with the inclusion of Hill’s coefficient *k* in the description. The qualitative nature of the results shown in this paper does not change with the values of *k*. Therefore, we have focused on exploring the effects of the remaining three parameters in the paper.

(14) The new discussion is quite nice as it relates the model to specific system behavior and predictions.

We thank the reviewer for the positive feedback and valuable comments.